# Lessons from Tailings Dam Failures—Where to Go from Here?

**David John Williams**

Geotechnical Engineering Center, School of Civil Engineering, The University of Queensland, Brisbane, QLD 4066, Australia; D.Williams@uq.edu.au

**Abstract:** Different regions worldwide have adopted various approaches to tailings management, as a result of the site settings and local practices as they have evolved. Tailings dam failures have continued to occur in both developing and developed countries, necessitating a range of tailings management approaches. These failures, while rare, continue to occur at a frequency that exceeds both industry and society expectations, and there is much to be learned from well-documented cases. Tailings management continues to be overly reliant on a net present value approach using a high discount factor, rather than a whole-of-life approach that may result in safer and more stable tailings facilities and may also facilitate the eventual mine closure. There is a need for the further development and implementation of new tailings management technologies and innovations, and for the application of whole-of-life costing of tailings facilities. Changes in tailings management will most readily be achieved at new mining projects, making change across the minerals industry a generational process.

**Keywords:** foundation failure; improved management; lessons; liquefaction; site settings; slurry tailings; tailings dam failures

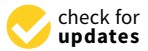

## 1. Introduction

The conventional deposition of slurry tailings behind dams that are raised progressively has led to an unacceptably high rate of catastrophic tailings dam failures, resulting in fatalities, damage to infrastructure and environmental harm. The rate of tailings dam failures is about two orders of magnitude higher than that of water dams, which are generally raised in a single lift to provide the required water storage.

Tailings dam failures that lead to high fatalities can prompt dramatic change, such as occurred in Chile, following the catastrophic El Cobre tailings dam failure during an earthquake. This change was initiated by tailings professionals, who questioned whether stable tailings dams could be constructed and operated in their highly seismic setting, and who made changes that ensured this was possible. The Global Industry Standard on Tailings Management (GISTM) and accompanying guides transpired as a result of the catastrophic tailings dam failure near Brumadinho in Brazil in January 2019. For change to be effective, active recognition and engagement is needed by all stakeholders.

This paper reviews the drivers of conventional tailings management, including the dominance of the net present value accounting approach, and the importance of each site's climatic, topographic and seismic settings that must be accommodated both during tailings operations and post-closure. Selected tailings dam failures are used to illustrate the main modes of failure of conventional tailings dams, and how they may be avoided, and the "as low as reasonably practicable" approach to reducing risk is discussed. The key features and requirements of the GISTM and the accompanying guides are reviewed, as is a comprehensive range of alternatives to the conventional deposition of slurry tailings behind dams. From these discussions, the expectations and implications of the GISTM, and the range of tailings management alternatives available, a path forward in tailings management is offered.

## 2. Conventional Tailings Management

### 2.1. Drivers of Conventional Tailings Management

The commonly held perception, supported by net present value (NPV) accounting, is that transporting tailings as a slurry to a surface dam is the most economical solution. Discounted long-term and rehabilitation costs "become" insignificant. Filtering tailings is perceived to be too expensive, despite reducing the storage volume and making it easier to rehabilitate to a high level. This has led to the widespread adoption of surface tailings facilities to store slurried tailings, delivered by robust and inexpensive centrifugal pumps and pipelines. Initially, small storage areas are constructed, leading to soft and wet tailings deposits, storing entrained water.

Operating costs increase over time as capital expenditure is avoided or delayed. This results in unintended increased storage volumes that must accommodate the entrained water, and more difficult rehabilitation of the tailings facility.

### 2.2. Importance of Site Settings

The key settings of a given mine site are the climate, the topography and the seismicity of the site. The site rainfall/precipitation and evaporation dictate the potential for the tailings to be exposed to desiccation. A dry climate makes slurry tailings disposal easier (e.g., in semi-arid regions of Australia, South Africa, and Southwestern USA). A wet climate has the potential to maintain the wetness of tailings (e.g., in the wet tropics, including Brazil). A near-neutral water balance can be tipped over into net positive by tailings deposition, or net negative post-closure by evaporation from stored water (e.g., for Canadian oil sands tailings).

The topography of the site dictates the volume of "free storage" available in valleys and the tailings dam's height. High seismicity will often govern tailings dam/storage design (e.g., in Chile and Peru). High seismicity may well need to be considered post-closure (in perpetuity) everywhere.

## 3. Lessons from Tailings Dam Failures

### 3.1. Selected Tailings Dam Failures

For the purposes of this paper, a number of tailings dam failures have been selected, as shown on the timeline in Figure 1 [1–12].

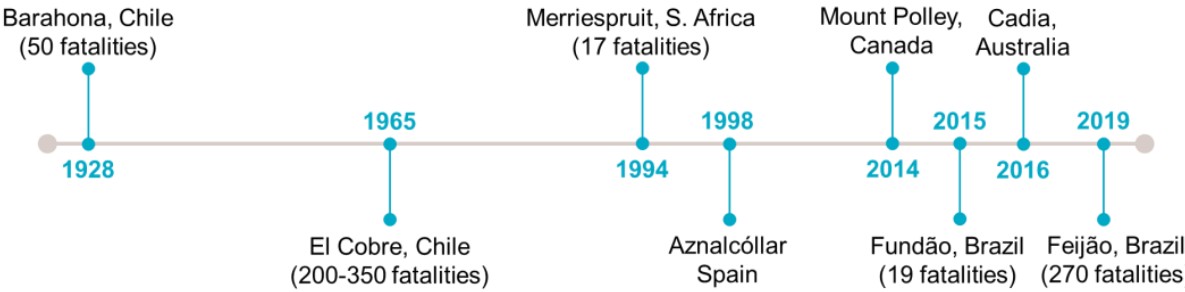

**Figure 1.** Timeline of selected tailings dam failures.

### 3.2. Some Root Causes of Tailings Dam Failures

Tailings dams that fail are marginally stable. Most tailings dams are stable and do not fail. Failures typically arise from a combination of causes, with water being a key element. Upstream constructed tailings dams are more prone to failure than downstream dams. Tailings deposited as a slurry are potentially susceptible to seismic or flow liquefaction, unless compacted and/or consolidated and desiccated. Weak foundation layers may cause tailings dam failures. Excessive tailings dam failures threaten the mineral industry's financial and social licenses to operate, and the control of their operations.

### 3.3. Seismic Liquefaction Failures of Upstream Sand Dams in Chile

*"Construction with, or on, liquefiable materials should be avoided"*, according to Dr Izzat Idriss [13]. *"You would not build a water dam using liquefiable materials or on a liquefiable foundation, so why would we do this for tailings dams?"* This is not to say that the upstream raising of a tailings dam cannot be made to work.

Among the fatal failures of steep upstream sand tailings dams in Chile, due to seismic liquefaction, are the Barahona No. 1 tailings dam failure in 1928 (Figure 2a) [1], and the El Cobre old tailings dam failure in 1965 (Figure 2b) [2]. The Barahona tailings dam underwent liquefaction due to a magnitude 8.2 earthquake, resulting in 50 fatalities, and did not lead to any changes to tailings management practices or regulations.

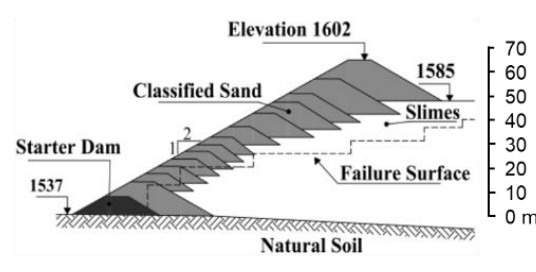
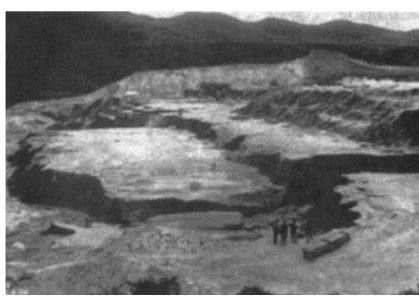

**(a)**

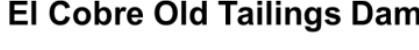
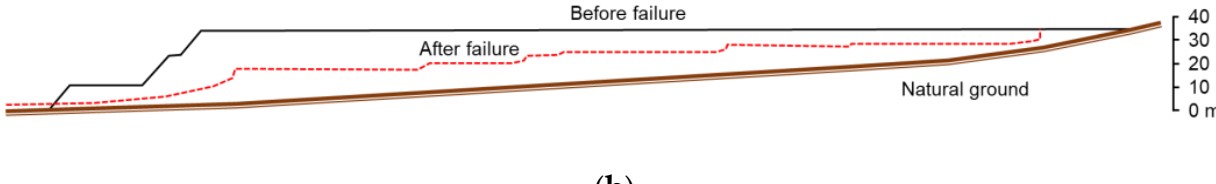

**(b)**

**Figure 2.** Failures of steep upstream sand tailings dams in Chile due to seismic liquefaction: (**a**) Barahona No. 1 tailings dam [1], and (**b**) El Cobre old tailings dam [2].

The El Cobre tailings dam also underwent liquefaction due to a magnitude 7.4 earthquake, inundating the town of El Cobre, resulting in 200 to 350 fatalities (the number is uncertain, due to the unknown number of occupants of the town at the time of the failure, and an inability to recover all of the bodies).

The El Cobre failure led to tailings practitioners in Chile questioning whether a safe tailings dam could be built to sustain large earthquakes. The industry responded rapidly by flattening and compacting the downstream slopes of sand dams from the natural 2 (horizontal) to 1 (vertical angle) to 4 to 1 (essentially a doubling of the margin of stability), and a change to the downstream construction, using either sand or earth fill. Later, a geomembrane was added to the exposed upstream face of the dam to limit the ingress of water from the contained slimes into the dam, and, in the case of sand dams, central sand cyclone stations replaced the series of cyclones along the crest of the dam. Central cyclone stations allowed greater control of the fines content of the sand (limiting this to less than 20%) and avoided weak spots being formed in the dam if one of the crest cyclones failed. Regulations to formalize these changes were implemented 5 years later and beyond, and these have also been adopted for tailings dams in seismically active Peru.

Typical cross-sections of downstream sand dams in Chile post-1965 are shown for the Las Tórtalas tailings dam (Figure 3) and the Quillayes tailings dam (Figure 4) [3].

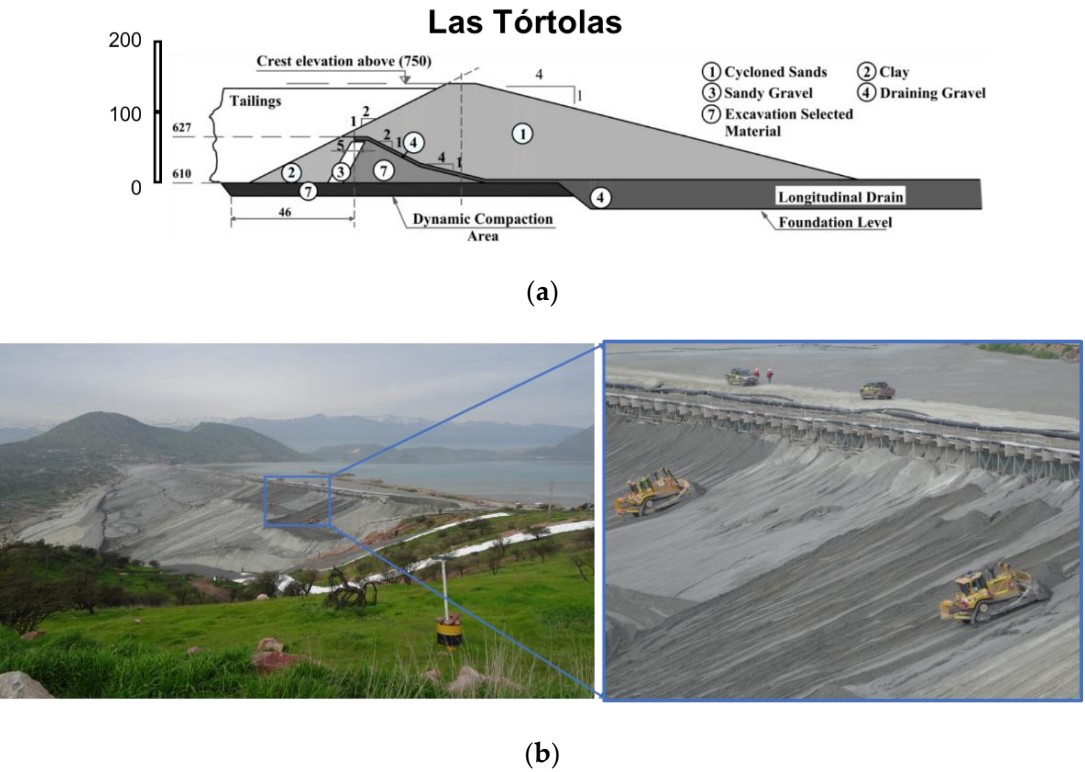

(**a**)

(**b**)

**Figure 3.** Las Tórtalas downstream sand dam [3]: (**a**) typical cross-section, and (**b**) flattening and compacting the downstream slope.

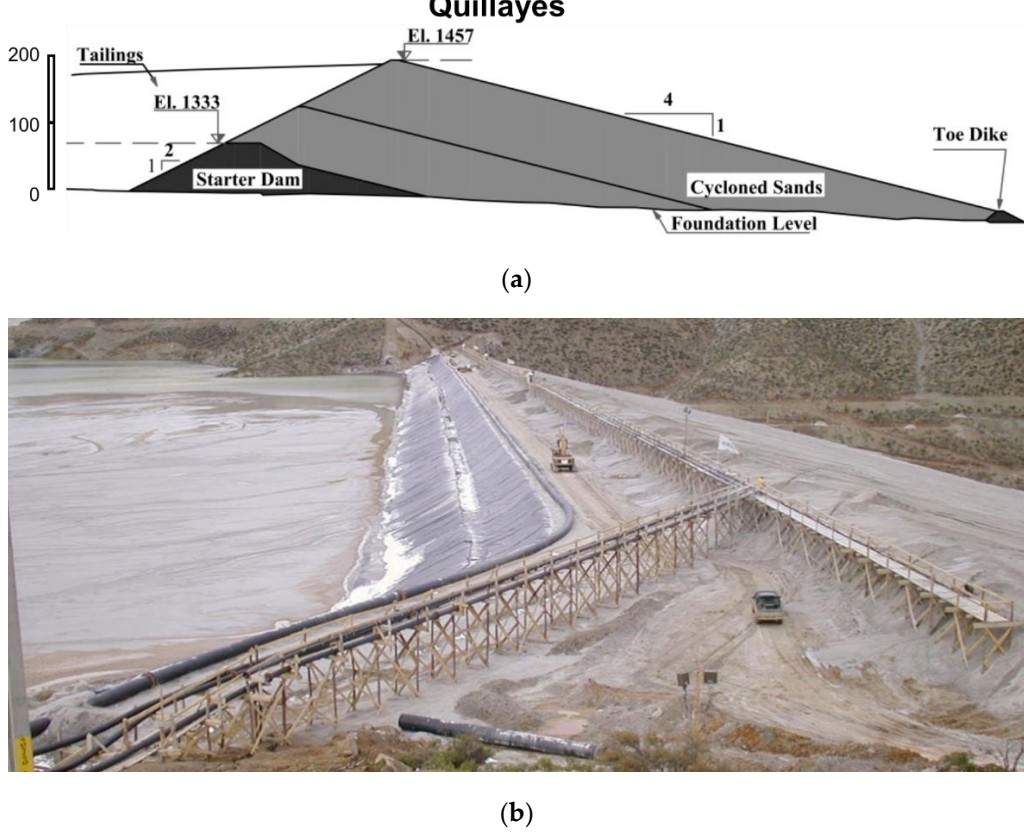

(**a**)

(**b**)

**Figure 4.** Quillayes downstream sand dam [3]: (**a**) typical cross-section, and (**b**) upstream geomembrane.

Of the approximately 740 tailings dams in Chile, about 100 are currently active, and these are mostly downstream sand dams. Around 470 tailings dams are inactive, and these are mostly upstream sand dams; about 170 are abandoned and are also mostly upstream sand dams. Active downstream sand dams in Chile have performed well since 1965, due to improved construction methods. Only one dam has failed, after the 8.4magnitude earthquake in 2010, resulting in four fatalities.

With the exception of a few inactive upstream sand dams in central Chile, most inactive and abandoned upstream sand dams have performed well, since they have drained down in the dry Chilean climate. This is a "good" story about the capability of the minerals industry in Chile, and later in Peru, to allow for the high seismicity, which governs the stability of tailings dams in their region.

### 3.4. Overtopping/Flow Liquefaction Failures of a Tailings Dam

The failure of the Merriespruit tailings dam in South Africa in 1994 followed a rainfall event that led to overtopping and erosion of the dam, due to its having inadequate free-board. The resulting flow liquefaction of the tailings engulfed the village of Merriespruit downstream, resulting in 17 fatalities [4], as depicted in Figure 5 [5].

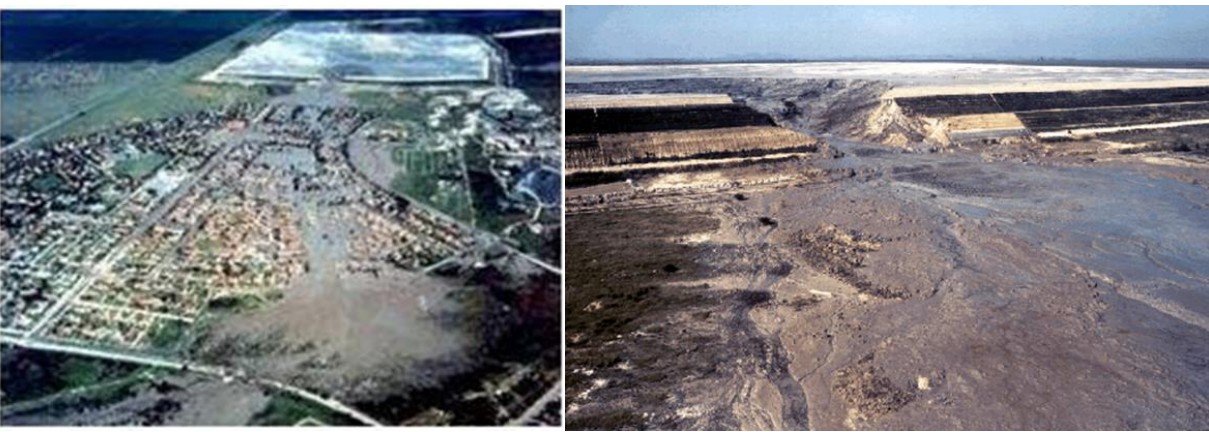

**Figure 5.** Failure of the Merriespruit tailings dam in South Africa, due to overtopping and flow liquefaction [5].

### 3.5. Limits to the Upstream Raising of a Tailings Dam

Upstream raising of a tailings dam that is partially on tailings relies on the tailings beach achieving sufficient consolidation and desiccation to form an adequate foundation for the raises. The higher the tailings dam constructed upstream, the further it extends over earlier tailings beaches, and the more likely it is that it will extend over earlier inundated tailings that may not provide an adequate foundation, as shown schematically in Figure 6. Hence, there are limits to how high the upstream raising of a tailings dam can be continued without having a wet and soft tailings layer at depth.

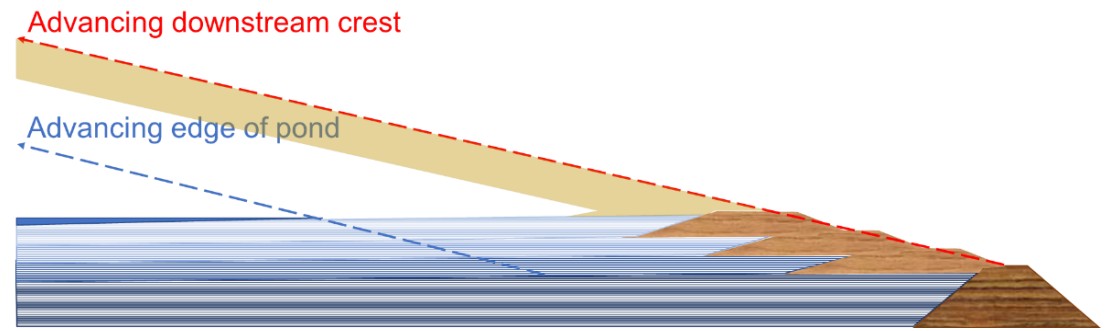

**Figure 6.** Schematic of the upstream raising of a tailings dam.

### 3.6. Limits to the Raising of a Tailings Dam on Weak Foundation Layers

As the height of a tailings dam is progressively raised, the pre-consolidation pressure (highest historical stress) of the foundation layers may eventually be exceeded. When this occurs, the foundation layer reverts from over-consolidated to normally consolidated. The stiffness of the now normally consolidated layer is perhaps a tenth of the previously over-consolidated layer; hence, deformations under additional loading are now 10 times higher. Such deformations can threaten the dam freeboard, potentially leading to overtopping and failure of the tailings dam. Examples of foundation failures of tailings dams include Aznalcóllar in Spain in 1998 [6], Mount Polley in Canada in 2014 [7], and Cadia in Australia in 2016 [8], as shown in Figure 7.

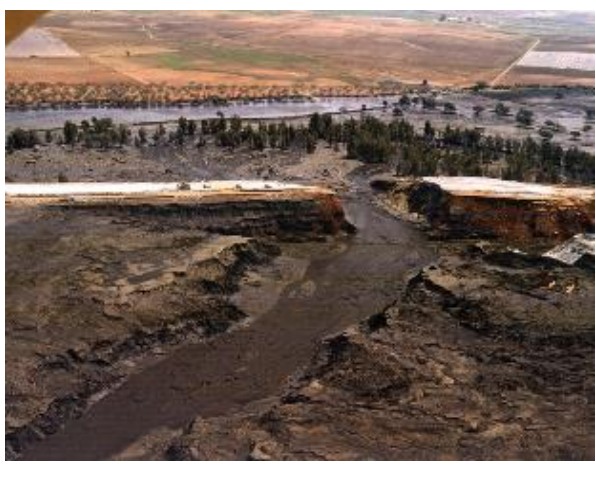

(**a**)

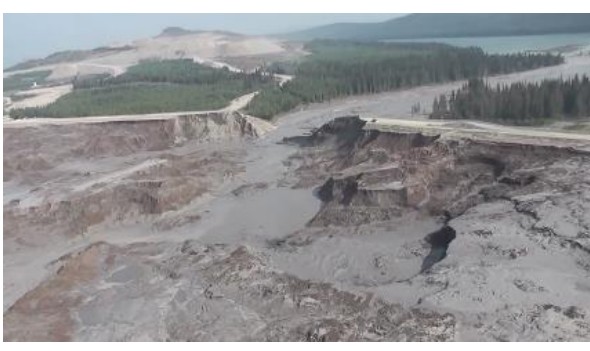

(**b**)

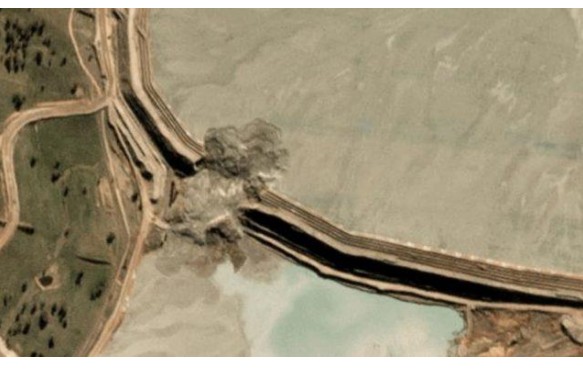

(**c**)

**Figure 7.** Failure of tailings dams due to weak foundation layers: (**a**) Aznalcóllar [6], (**b**) Mount Polley [7], and (**c**) Cadia [8].

### *3.7. Rapid Flow Liquefaction of Upstream Tailings Dams*

In Minas Gerais, Brazil, the Fundão tailings dam failure in November 2015 caused 19 fatalities and extensive infrastructure and environmental damage (Figure 8) [9], and the Feijão tailings Dam I failure near Brumadinho in January 2019 caused 270 fatalities and infrastructure and environmental damage (Figure 9) [10]. Both incidents were rapid flow liquefaction failures.

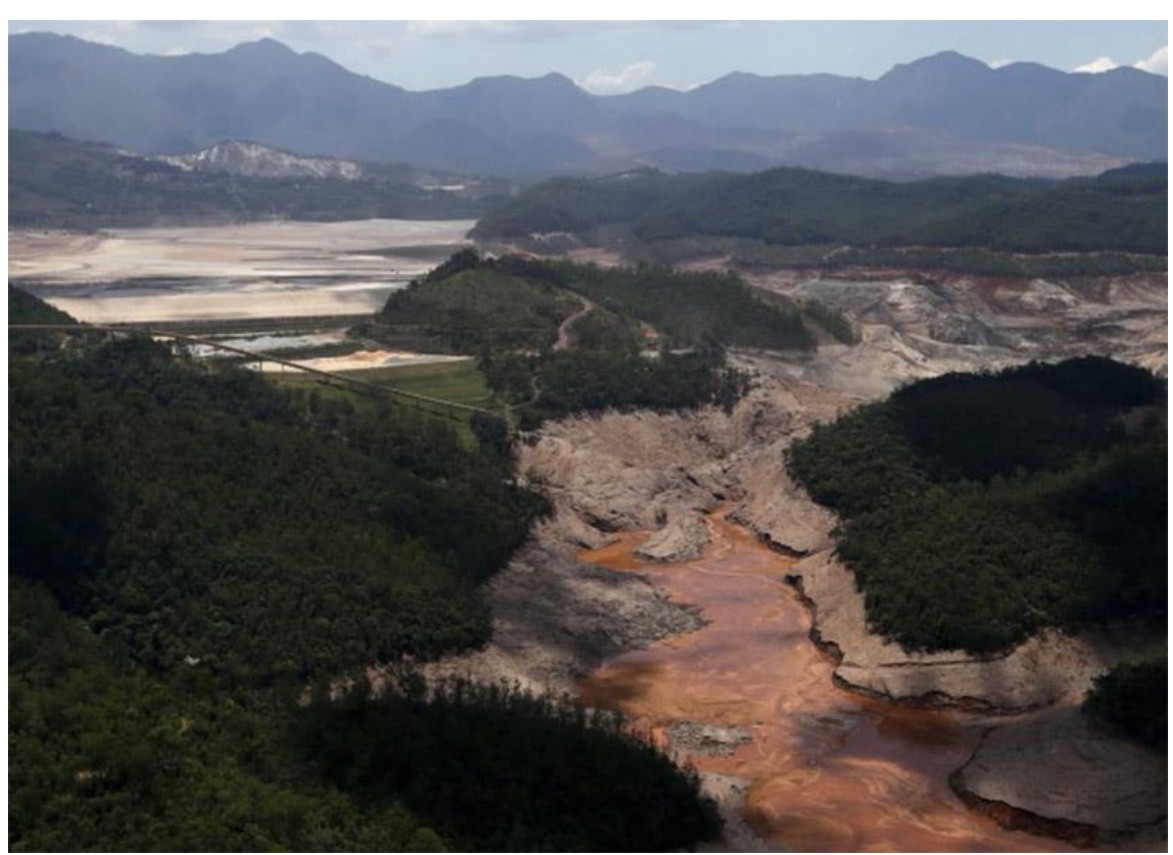

**Figure 8.** Failure of the Fundão tailings dam due to rapid flow liquefaction [9].

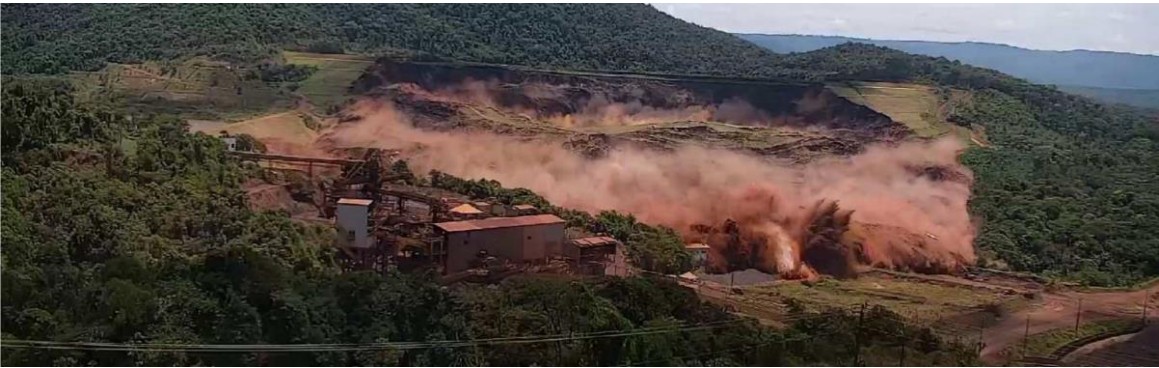

**Figure 9.** Failure of Feijão tailings dam I due to rapid flow liquefaction [10].

Based on the findings in [11], the technical causes of the Fundão tailings dam failure may be summarized as follows. Regarding the question of why flow liquefaction occurred, damage to the original starter dam resulted in increased saturation of the tailings, and slimes were deposited very rapidly and closer to the dam than the designer intended. The failure of a concrete drain and a setback of the left abutment caused the dam to be

raised over the slimes, and compression and lateral extrusion of the slimes resulted in the loosening of sand layers above. Regarding the question of why the flow liquefaction occurred where it did, the setback left abutment, on softer, wetter slimes, was critical to stability. Regarding the question of why the flow liquefaction occurred when it did, a series of three small seismic shocks occurred about 90 min prior to the failure, resulting in a small deformation that was sufficient to initiate liquefaction of the marginally stable setback.

Based on [12], the tailings in the Feijão tailings Dam I had high iron content, and oxidation of the iron on exposure to the air led to bonding, rendering the tailings stiff and brittle, with the potential for significant and rapid strength loss with ongoing strain on undrained loading. It is noted that a limit equilibrium stability analysis assumes all points on a slip surface to be at the same state, which is not necessarily correct for a marginally stable dam, in which some material may be pre-peak strength, some at peak strength, and some post-peak.

Dam I was marginally stable. The upstream slope was over-steep. A setback pushed the upper raises of the dam over weaker fine tailings. The water level in the dam remained persistently high, due to a lack of sufficient internal drainage. The bonded tailings had the potential for very brittle behavior if triggered to become undrained. The dam was subjected to high and intense wet season rainfall that can result in a significant loss of suction in the unsaturated tailings above the water level, producing a small loss of strength.

It was concluded in [12] *"that the sudden strength loss and resulting failure of this marginally stable dam was due to a critical combination of ongoing internal strains due to creep, and a strength reduction due to loss of suction in the unsaturated zone caused by intense rainfall towards the end of 2018. This followed a number of years of increasing rainfall and intensity after tailings deposition ceased in July 2016. Calculated pre-failure strains from this combination of triggers matched well the small deformations of Dam I in the year prior to the failure."*

A schematic of the path to failure of Dam I is shown in Figure 10, which shows a plot of undrained strength to vertical effective stress ratio against vertical strain. Different points within the dam would be at different starting points on the stress/strain plot, shown in Figure 10.

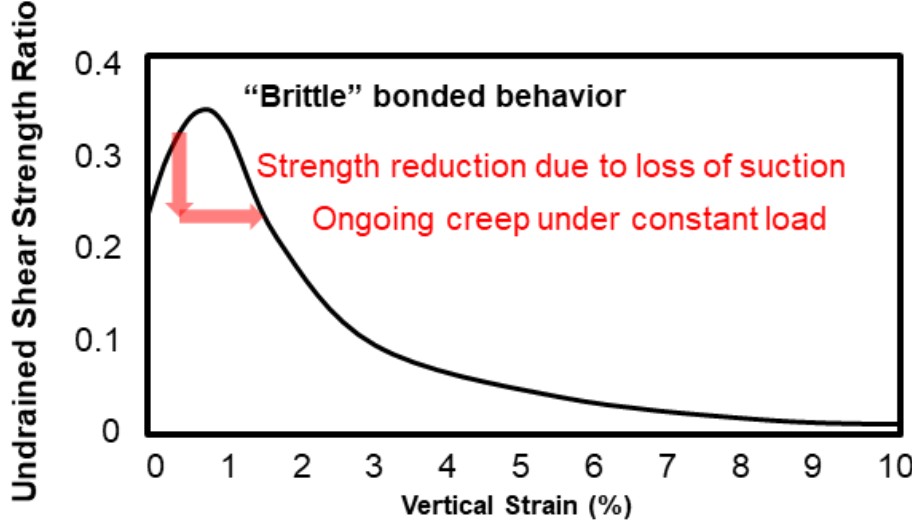

**Figure 10.** Schematic of the path to failure of Dam I.

## 4. As Low as Reasonably Practicable

The "as low as reasonably practicable" (ALARP) principle arose out of the UK Health and Safety at Work Act 1974 [14], and this is increasingly applied to managing tailings dam risk, among others. A typical schematic of the ALARP principle is shown in Figure 11 [14], which includes the probabilities of fatalities for workers and the public who are exposed to the risk ($1 \times 10^{-3}$ and $1 \times 10^{-4}$, respectively) and for the public generally ($1 \times 10^{-6}$).

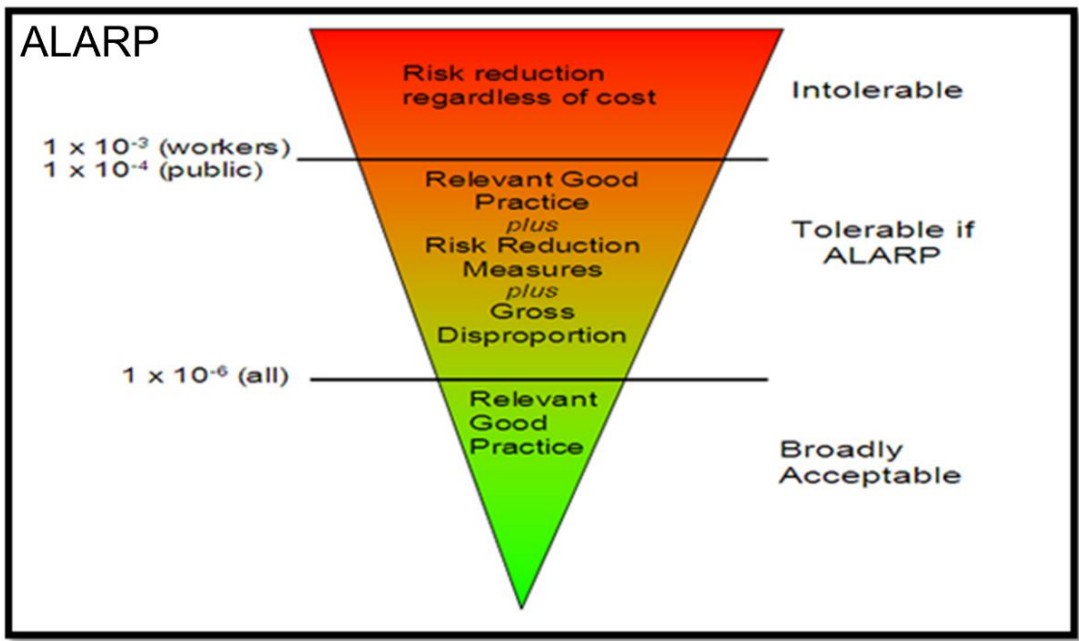

**Figure 11.** ALARP principle, including the probabilities of fatalities [14].

The ALARP principle can be applied in the risk assessment process, involving the frequency of occurrence versus the magnitude of consequence, as described in [15] and shown schematically in Figure 12. The aim is to carry out risk mitigations that either reduce the frequency of occurrence (F) or reduce the magnitude of consequence (N), or both, to shift the risk from the "intolerable" region into the "tolerable" region.

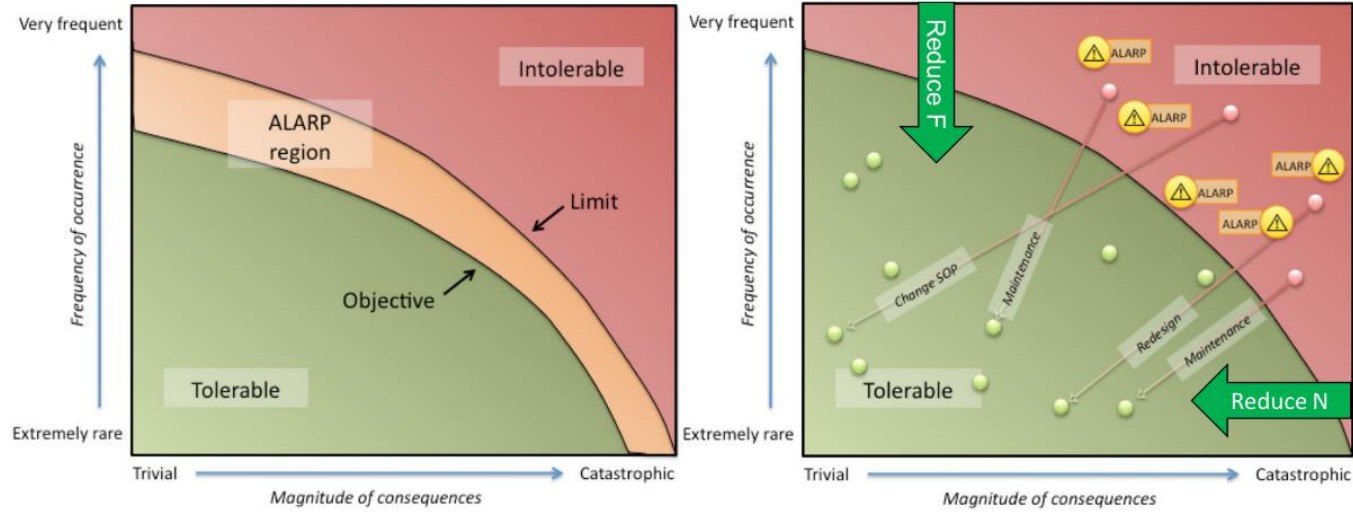

**Figure 12.** Schematic of ALARP principle applied in risk assessment [15].

In engineering risk assessment, the plot showing the frequency of occurrence versus the magnitude of consequence becomes a plot of annual probability of up to N fatalities (F) versus the number of fatalities (N). Some examples of actual F–N data for different engineering infrastructures are shown in Figure 13, after [16]. Superimposed on Figure 13 are the ALARP limits recommended by the Canadian Dam Association [17], shaded in red for additional risk, orange for tolerable risk, and green for broadly acceptable risk.

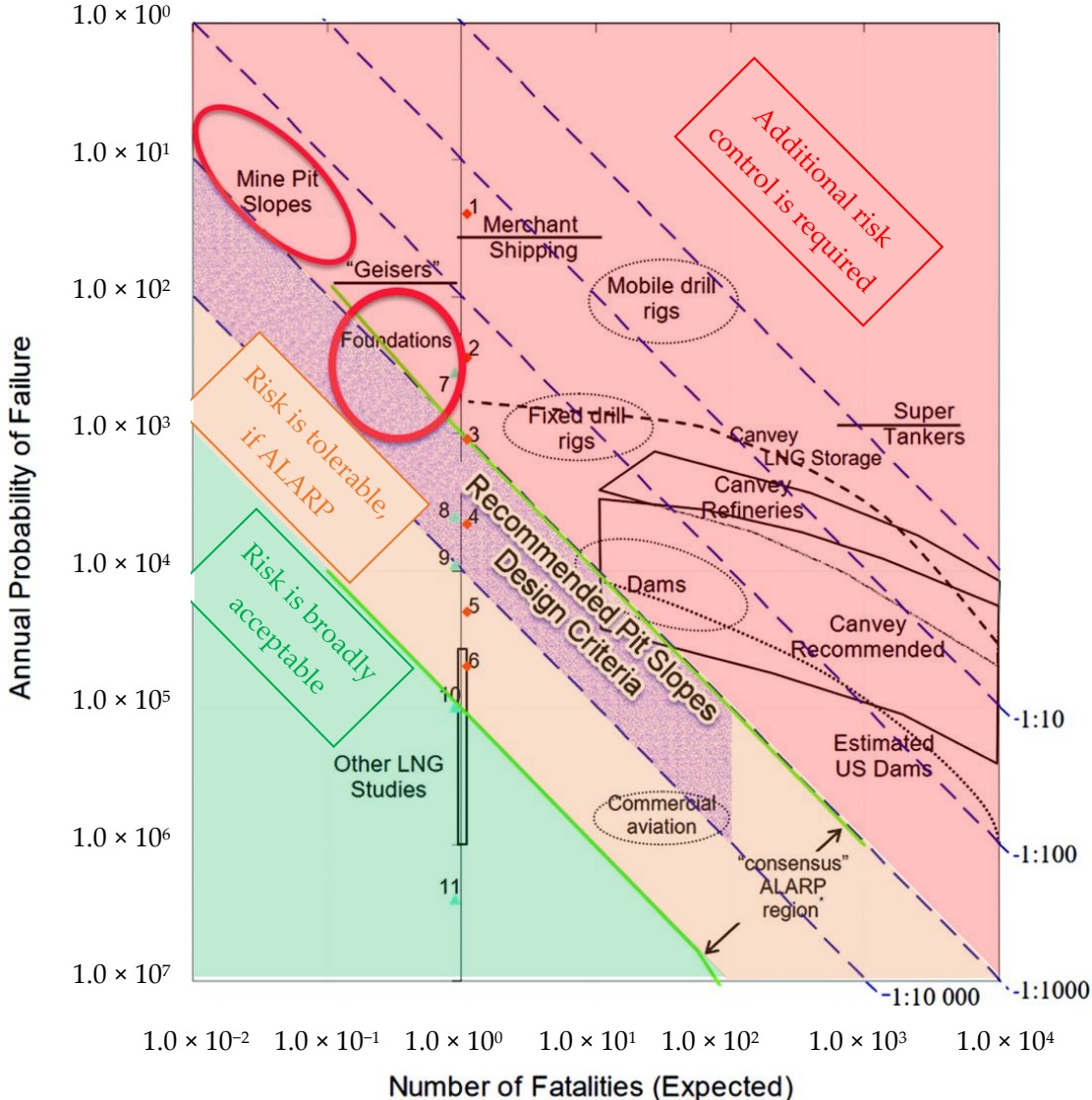

**Figure 13.** Examples of actual F–N data for different engineering infrastructure [16,17].

\* region defined by the upper most and lower most ALARP boundaries from: Hong Kong planning department, ANCOLD, U.S. Bureau of reclamation and U.K. HS executive guidelines.

**Individual Fatality statistics U.S.A - Voluntary**

1. Space Shuttle program (per flight)   2. Cigarette smoking (1 pack per day)   3. Average individual voluntary sporting risk
4. U.S. Police killed in line of duty (total)   5. Frequent flying profession   6. Alcohol (light drinking)

**Individual Fatality statistics U.S.A - Involuntary**

7. Cancer   8. Motor vehicle   9. Home accidents   10. Air Travel   11. Hurricanes / Lightning / Tornadoes

There is a mismatch between actual F–N data and the aspirational ALARP target range, let alone the "acceptable" range. It could be argued that the actual F–N data are "accepted", since they represent the status quo that we live with. "Acceptable" is a subjective term, with different people having different perceptions depending on their exposure to risk, whether this exposure is "involuntary" or "voluntary", their level of awareness of risk, and their sense of social responsibility.

It is clear from Figure 13 that water dams plot up to an order of magnitude above the ALARP region, warranting additional risk control. Tailings dams plot on average two orders of magnitude above water dams, warranting considerably more risk control than water dams. The *ICMM Tailings Management Good Practice Guide* [18] provides the ALARP schematic shown in Figure 14, with the "sweet spot" shown by the green circle representing the optimal balance between reduced risk for the resources and effort input.

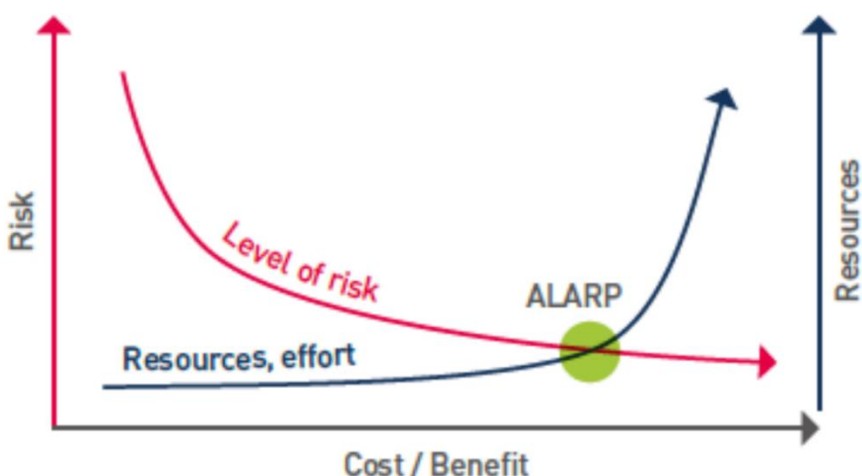

**Figure 14.** Schematic of the ALARP "sweet spot" [18].

## 5. Global Industry Standard on Tailings Management

The failure of the Feijão tailings Dam I led to the launch of the Investor Mining and Tailings Safety Initiative [19], which sought reassurance from the minerals industry about the safety of their tailings dams. The International Council of Mining and Metals (ICMM), the United Nations Environment Programme (UNEP), and the Principles for Responsible Investment (PRI), under the "Global Tailings Review", co-convened the development of a Global Tailings Standard under an independent chair and expert panel. The Global Industry Standard on Tailings Management (GISTM) was launched in August 2020 [20].

### 5.1. What Does the GISTM Require of Tailings Facility Operators?

The GISTM has the aspirational goal of *"zero harm to people and the environment"* from tailings facilities [20]. It elevates accountability to the highest organizational levels, with new requirements for independent oversight. It expects global transparency and disclosure to improve stakeholder understanding. It sets as the design basis an *"Extreme Consequence Classification external loading criteria or the current Classification, with upgrade to Extreme maintained throughout the tailings facility lifecycle"*. Many active tailings facilities are not classified as having an "Extreme" consequence classification and may need to be re-classified and/or upgraded to meet "Extreme".

The post-closure classification is "Extreme" since tailings facilities must remain safe, stable, and non-polluting in perpetuity. This criterion is in line with current tailings dam guidelines, such as in [17,21]. The extreme consequence return interval is 1 in 10,000 years, which is presumed to be equivalent to "in perpetuity". This long return interval is not applied to other engineered structures, which can be attributed to the loss of confidence and trust in the capability of the industry, their consultants, and regulators to ensure safe and stable tailings dams.

The purpose of the GISTM is to provide *"a framework for safe tailings facility management, while affording operators flexibility as to how best to achieve this goal"*, in other words, to be self-regulating. The design, construction, operation, monitoring, and closure of tailings facilities are required to be robust in order to minimize the risk of harm to people and the environment. The GISTM calls for emergency response and long-term recovery of the community affected, in the event of a tailings facility failure. Public disclosure and access to information are required to support public accountability, requiring clear communication. The GISTM also specifies governance requirements for the management of tailings facilities.

### 5.2. What Are the Next Steps in the Implementation of the GISTM?

Since the tailings dam failure near Brumadinho in Brazil, some regions of the world, now including Brazil, have been refused insurance for tailings dams. Other regions have had their insurance coverage reduced (by about one-third, from coverage that was

typically USD 300 million to USD 200 million, which is clearly insufficient to cover a catastrophic failure), and their premiums have increased (about two-fold). Some major mining companies are self-insuring their tailings facilities. However, mid-tier and small mining companies with few mines cannot afford to not be insured. Access to finance for new tailings facilities could be withheld from companies that do not comply with the GISTM.

The Global Tailings Review estimated that there are approximately 18,000 tailings facilities worldwide, of which approximately 3500 are active. The GISTM does not cover the thousands of inactive and abandoned tailings facilities worldwide.

ICMM members, comprising 27 of the world's major mining and metals companies and operating about 1200 tailings facilities (about one-third of the world's active facilities), have committed that all tailings facilities with "Extreme" and "Very high" potential consequences will be in conformance with the GISTM by August 2023, and all other facilities in conformance by August 2025 [20]. Non-ICMM members companies are encouraged to comply with the GISTM.

### 5.3. Management of Tailings Facilities

The ICMM and its members are adapting the management structure and interactions of their "Extreme" and "Very high" consequence classification tailings facilities to accommodate the governance requirements of the GISTM. The new structure and interactions are along the lines of those shown schematically in Figure 15. The GISTM defines the roles and responsibilities of all personnel in this structure. The Accountable Executive is directly answerable to the Chief Executive Officer (CEO) on matters related to the GISTM, and will communicate with the Board of Directors (Board). The Accountable Executive is accountable for the safety of the tailings facility, and for minimizing the social and environmental consequences of a potential tailings facility failure. The Accountable Executive interacts closely with the Tailings Facility Operator, who in turn interacts closely with the Responsible Tailings Facility Engineer (RTFE). The Accountable Executive should not have Key Performance Indicators (KPIs) that are based on production improvements.

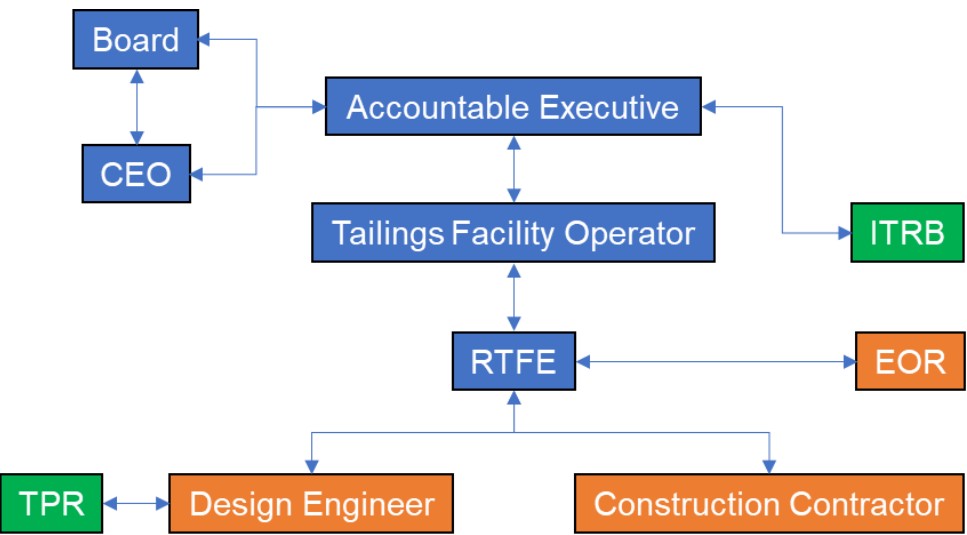

**Figure 15.** Schematic tailings facility management structure and interaction.

The RTFE works closely with the (usually) external Engineer of Record (EOR), typically interacting monthly, meeting on-site perhaps quarterly, and with the two having the authority between them to make recommendations to the Accountable Executive about the ongoing operation of the tailings facility. The EOR has responsibility for preparing and updating the Design Basis Report that provides a basis for the monitoring and risk management of all design phases of the tailings facility.

The RTFE and EOR also work closely with the Design Engineer and Construction Contractor for the tailings facility. The design of the facility and sequential raises are reviewed by a third-party reviewer (TPR). According to the GISTM [20], dam safety reviews are periodically and systematically carried out by a TPR *"to assess and evaluate the safety of a tailings facility against* ["credible"] *failure modes, in order to make a statement on the safety of the facility. A safe tailings facility is one that performs its intended function under both normal and unusual conditions, does not impose an unacceptable risk to people, property or the environment, and meets applicable safety criteria."*

Credible failure modes are "*technically feasible failure mechanisms given the tailings and foundation, their parameters, geometry, drainage conditions and surface water control, throughout the lifecycle of the tailings facility. They typically vary during the lifecycle of the facility as the conditions vary, and the tailings facility requires sufficient resilience against each credible failure mode. Different failure modes will result in different failure scenarios. Credible catastrophic failure modes do not exist for all tailings facilities, and are not associated with likelihood and not a reflection of facility safety.*"

According to the GISTM [20], the Independent Tailings Review Board (ITRB) "*provides independent technical review of the design, construction, operation, closure and management of the tailings facility. ITRB members are third parties who are not, and have not been, directly involved with the design or operation of the particular tailings facility. The expertise of the ITRB members should reflect the range of issues relevant to the facility, and its context and the complexity of these issues.*"

An ITRB would typically include a geotechnical expert, a hydrology or hydrogeology expert, as appropriate, a seismology expert for highly seismic locations, and possibly a geochemistry or geomorphology expert, as appropriate. The ITRB would typically meet prior to key changes in the life of the tailings facility, such as a dam raise, interacting most closely with the Tailings Facility Operator and the RTFE, and also with the EOR and Design Engineer, and possibly also with the TPR. The ITRB should report its findings through the Accountable Executive to inform the company's CEO and board.

A qualified EOR, TPR, or ITRB member would typically be able to handle perhaps four tailings facilities. An estimated 70 ITRBs would be required for all operating tailings facilities globally, requiring over 200 tailings experts, with perhaps 10 times this number of EORs and TPRs required. There are insufficient tailings experts available worldwide to meet this demand, particularly in the current COVID-19 climate that restricts travel to sites. The increasing availability of online training and certification in tailings management will assist in meeting this demand in the future, although the experience required for these roles will take time to develop. Senior tailings professionals will need to mentor talented early-career professionals.

*5.4. ICMM Documents Accompanying the GISTM*

With the release by the ICMM of the *Tailings Management Good Practice Guide* [18], and conformance protocols [22], comes a structure for tailings management, as shown in Figure 16. At its base are technical guidelines from several institutions: the International Committee on Large Dams (ICOLD, which is currently working toward a global tailings guideline); the Australian National Committee on Large Dams (ANCOLD) [21]; the Canadian Dam Association (CDA) [17]; the Japanese Commission on Large Dams (JCOLD); and the South African National Committee on Large Dams (SANCOLD). The United States Society on Dams is drafting a national tailings dam guideline to replace the individual state guidelines, many of which were drafted in the 1990s and lack consistency. See [23] for a review of mine tailings guidelines, initiatives and standards.

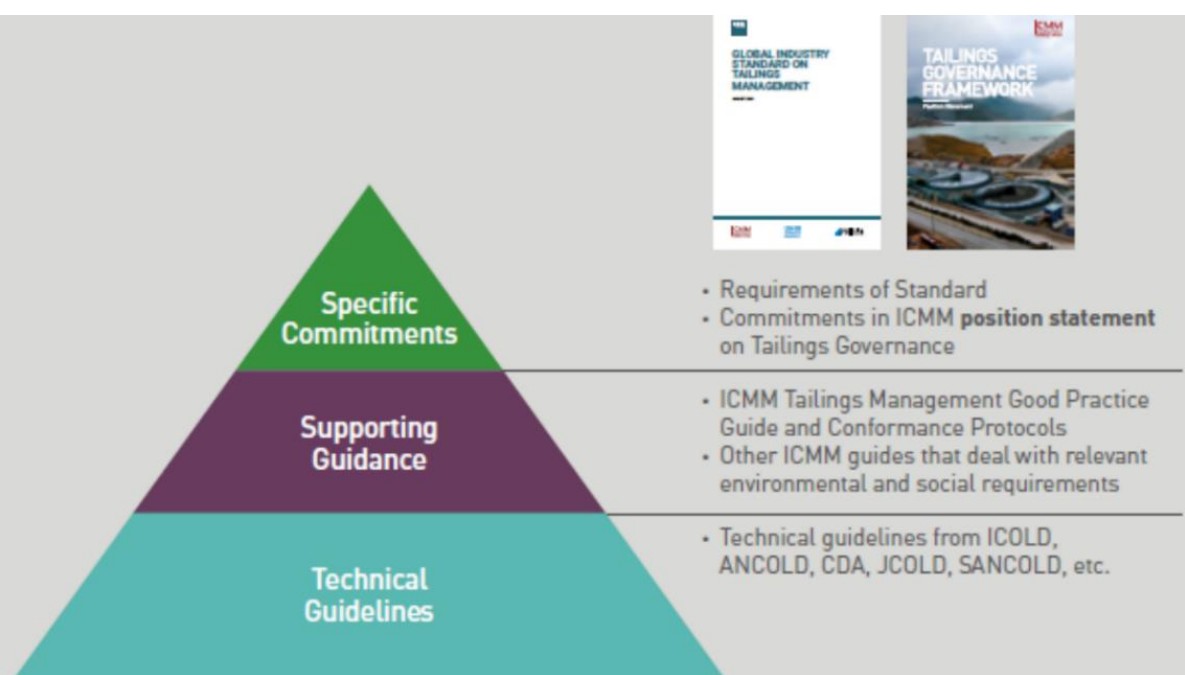

**Figure 16.** Structure for tailings management [18].

At the apex of the structure shown in Figure 15 are the requirements of the GISTM and commitments in the ICMM position statement on tailings governance [24]. Supporting guidance linking the technical guidelines and the specific commitments includes the *ICMM Tailings Management Good Practice Guide* and conformance protocols, plus other ICMM guides that deal with relevant environmental and social requirements.

It remains to be seen how this structure will be accommodated within the regulation of tailings facilities, both regionally and globally. None of the structure shown in Figure 15 is specifically included in the legislation, although in the jurisdictions where they are applied, the ANCOLD [21] and CDA [17] guidelines are taken by mining companies, regulators, and the courts to be de facto "standards" for tailings dam design.

## 6. Alternative Tailings Management Options

*"In the wake of recent tailings dam failures, improved Management Resilience (that is, Governance) is leading over improved Engineering Resilience, which would reduce Liability"*, according to Dr. Andy Robertson [25].

The GISTM [20] does not specify particular tailings management options that should be allowed or disallowed, leaving this choice to the Operator. However, the GISTM does require that tailings management options be reviewed and analyzed. For existing tailings facilities, the Operator is required *"to periodically review and refine the tailings technologies and design, and management strategies, to minimize risk and improve environmental outcomes. An exception applies to facilities that are demonstrated to be in a state of safe closure."* For new tailings facilities, the Operator is required *"to use the knowledge base and undertake a multi-criteria alternatives analysis of all feasible sites, technologies and strategies for tailings management. The goal of this analysis is to: (i) select an alternative that minimizes risks to people and the environment throughout the tailings facility lifecycle; and (ii) minimize the volume of tailings and water placed in external tailings facilities."*

The GISTM also requires the development and documentation of a *"breach (dam break and runout) analysis for the tailings facility using a methodology that considers credible failure modes, site conditions, and the parameters of the tailings. When flowable materials (water and liquefiable solids) are present at tailings facilities with a Consequence Classification of 'High', 'Very High' or 'Extreme', the breach analysis results should include estimates of the physical area impacted by a potential failure, flow arrival times, depth and velocities, and depth of material deposition."*

The results of the breach analysis should be used to identify and document the potential human exposure and vulnerability to credible failure scenarios regarding a particular tailings facility. The breach analysis should be updated whenever there is a material change either to the tailings facility or the physical area impacted.

According to Williams [26], *"while the conventional disposal of tailings slurry can be the optimal NPV and life-cycle choice for a given operation, there is often a divergence when a whole-of-life approach is fully considered."* Alternative approaches to tailings management are described in the following sections.

### 6.1. Geotechnical versus Geochemical Stability of Stored Tailings

Fundamentally, the geotechnical stability of tailings is enhanced by dewatering and densification, while the geochemical stability of potentially contaminating tailings (such as those containing sulfides) is ensured by maintaining the tailings at near-saturation, to limit oxygen ingress. For the storage of tailings slurry in a surface facility, geotechnical stability and the potential for upstream construction are enhanced by cycling tailings deposition in thin layers, and by leaving time for consolidation and desiccation before the next layer is deposited.

Geochemical stability is ensured by maintaining the tailings at a level that is at least 85% saturated, to limit the diffusion of oxygen into a desaturating surface. This can be achieved by depositing fresh tailings slurry before the previous layer has been allowed to desiccate to less than this saturation limit. Geochemical stability is also enhanced for fine-grained and clay mineral-rich tailings that tend to hold onto water due to their low permeability. Perhaps counterintuitively, sulfidic tailings that tend to hardpan result in a virtually impermeable crust that limits moisture and hence contaminant flow, either as net downward percolation or as evapotranspirative uptake into an overlying growth medium. Hardpans can have a permeability that is 10 to 100 times lower than that of uncemented tailings, and an oxygen diffusion rate up to 1000 times lower, causing them to maintain a high degree of saturation and to not pass water or oxygen [27]. However, hardpans may not be continuous across the tailings' surface.

For potentially contaminating tailings that are deposited in-pit, any pit lake should not be allowed to become a "source" of contaminated water for surface or ground waters. It is preferable that final pit lakes containing water of diminishing quality should be avoided by completely back-filling the pit, even if the pit were to remain a surface- and groundwater "sink".

### 6.2. In-Plant Dewatering of Tailings

The in-plant recovery of process water is the most effective means of maximizing water return for recycling and for recovering any residual process chemicals, assuming that the return water is suitable for reuse in the plant and that the tailings can be dewatered. The choice of the appropriate amount of in-plant dewatering is related to the tailings dewatering continuum adapted from [28], as shown in Figure 17, which ranges from the initial slurry-like state to a potentially soil-like state. The cost of mechanical dewatering of tailings increases exponentially with increasing dewatering, which is balanced by benefits including the greater recovery of water and process chemicals and reduced tailings storage volume.

Clay mineral-rich tailings, such as oil shale tailings, some coal tailings, and caustic processing residues, such as red mud from bauxite refining, nickel laterite tailings, and oil sands tailings, are difficult to thicken, let alone filter. Slurry and thickened tailings may readily be pumped using robust, inexpensive centrifugal pumps. Paste tailings require positive displacement or diaphragm pumps, which are an order of magnitude more expensive than centrifugal pumps and are more sensitive to variations in the input tailings particle size distribution and chemistry. Furthermore, a true paste will discharge like "toothpaste" and require continuous management of the deposition. To avoid the need to

pump paste tailings, they may be discharged through gravity to a completed underground mine, possibly as cemented paste tailings backfill, or to a completed pit.

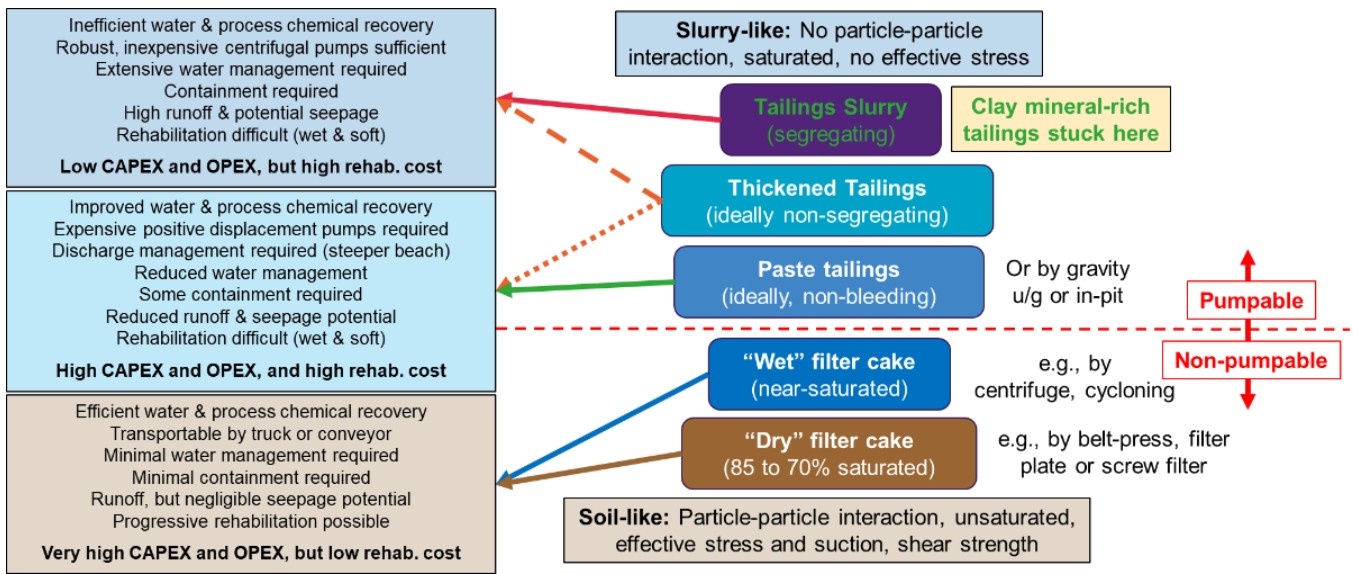

**Figure 17.** The tailings dewatering continuum [28].

"Wet" tailings filter cake, which is potentially produced by centrifuging or cycloning, is not pumpable, can potentially be conveyed or trucked, and typically flows on deposition. "Dry" tailings filter cake, which is potentially produced by a belt-press or filter plate, and possibly by screw filter (used in sewage treatment, although it has not been applied to tailings), can be conveyed, trucked, and potentially mixed with coarse-grained wastes, and will not flow on deposition.

The optimal in-plant dewatering of tailings for disposal to a surface tailings storage facility is likely to be of tailings thickened as much as possible but that is still able to be pumped by robust, inexpensive centrifugal pumps. Paste tailings are best reserved for gravity disposal underground or in-pit. Filtered tailings are potentially suited to "dry" stacking, possibly with the compaction of at least the outer zone to prevent potential liquefaction.

*6.3. In-Facility Dewatering of Tailings*

Water recovery from a surface tailings facility is generally limited to the recovery of supernatant water (the water that pools at the end of the tailings beach). Tailings water is lost to evaporation from the wet tailings and from the decant pond, from entrainment within the tailings, and from seepage into the foundation and through the dam.

While the majority of the water from a tailings facility may be lost through evaporation off freshly deposited wet tailings, minimizing the size of the decant pond and the rapid return of supernatant water will minimize additional evaporation losses. Tailings densification, due to self-weight consolidation, and desiccation on exposure, is far more robust than in-plant thickening or filtration, being less affected by variations in the nature of the tailings.

Water recovery from an in-pit tailings facility is more difficult than from a surface tailings facility. Firstly, the rate of rising in-pit is generally much faster, due to the steep geometry of the pit, compared with more shallow surface tailings facilities. Secondly, it is difficult to recover water from a pit as it fills. Thirdly, the opportunity for desiccation of in-pit tailings is limited by the usual water cover, and by shadows thrown from the pit walls onto any exposed tailings. The faster rate of rising in-pit, only partial consolidation,

and limited or absent desiccation, result in limited dewatering and densification of the tailings, and much higher water entrainment.

### 6.4. Coagulation, Flocculation and Secondary Flocculation

Coagulation concerns the initial supernatant water clarification through the settling of colloids. Coagulation is both a chemical (charge neutralization of colloids) and a physical (consequent aggregation of colloids) process. Coagulants are mostly cheap iron or aluminum salts, forming a highly ionic polymer of low molecular weight.

Flocculation concerns the agglomeration of dispersed particles. Flocculants are normally high molecular weight, long-chain polymers that achieve flocculation relatively slowly, primarily by bridging, but also by wrapping water. Polyacrylamide is a common flocculant, either as negatively charged anionic for mineral particles or positively charged cationic for organic particles, with divalent cations such as $Ca^{2+}$ and $Mg^{2+}$ aiding bridge formation. The effectiveness of flocculation is dominated by the clay mineral content and type of the tailings, and the salinity of the process water.

If conventional thickening and slurry tailings disposal fail to achieve adequate settling and consolidation and supernatant water recovery, secondary (inline) flocculation can be applied just prior to the discharge of the tailings, to re-flocculate conventionally thickened tailings that have been shear-thinned by pumping.

It is important that the appropriate type and dosage of coagulants and flocculants applied to tailings be carefully selected, which generally involves trialing them both in the laboratory and on-site. Flocculants, in particular, can be expensive, and over-dosing can be counter-productive, leading to excessive "wrapping" of both the tailings particles and entrained water. With variable tailings feeds, online monitoring of pH and electrical conductivity may be required, and the flocculant type and dosage should be adjusted to suit.

### 6.5. On-Off Tailings Cells

As described in Williams [26], the desiccation and harvesting of black coal tailings in "on-off" tailings cells have been employed at a number of mines, including at Charbon Coal Mine in New South Wales, Australia, since 1990, as shown in Figure 18. This method involves the disposal of slurry tailings in cells in thin layers, preferably no more than 600 mm thick, and leaving sufficient time for the tailings to consolidate and desiccate on exposure to the sun and the wind in dry weather, prior to the disposal of the next layer. The limited layer thickness is dictated by desiccation by solar and wind action, which drops off exponentially with depth. The time required for the consolidation and desiccation of each layer will be of the order of several weeks, depending on the nature of the tailings and the ambient weather conditions. The cells are filled to a depth that can readily and safely be excavated by the available equipment, such as an excavator (as shown in Figure 18a for poorly desiccated tailings) or a front-end loader (as shown in Figure 18b for well-desiccated tailings). A typical depth is about 3 m. The full depth of dried tailings is then harvested and can be co-deposited with coarse wastes (as shown in Figure 18c). This method of dewatering tailings requires a large number of shallow cells covering a large footprint, although probably no larger a footprint than would ultimately be needed for a conventional surface slurry tailings facility. Reliance on evaporation means that a considerable volume of water is lost. Ultimately, there could potentially be no tailings facility remaining. The cost of operating on-off tailings cells is comparable to the cost of tailings filtration and dry stacking.

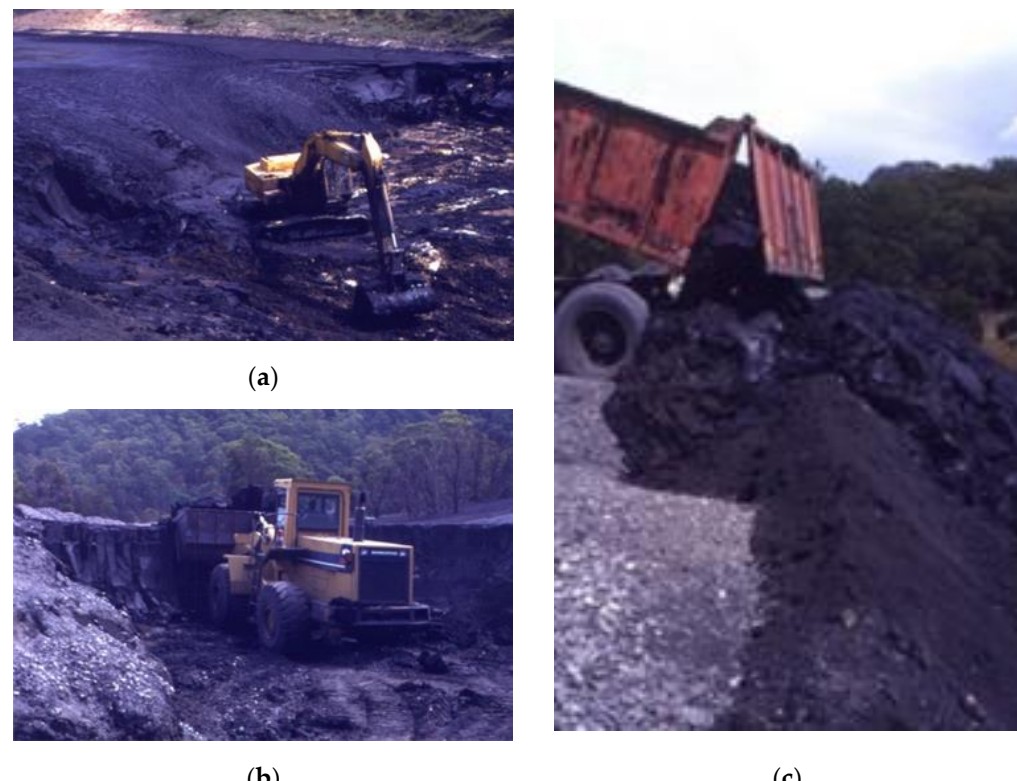

**Figure 18.** On-off coal tailings cells, showing: (**a**) insufficient desiccation, (**b**) effective, full-depth desiccation, and (**c**) dumping harvested tailings with the coarse reject.

*6.6. Farming of Tailings*

Some forms of wet and soft tailings, particularly clay mineral-rich tailings and caustic process residues, may benefit from "farming" by the use of equipment, such as an amphirol or scroller (being the more generic term), and/or later compaction by a D6 Swamp Dozer. Farming is widely applied to red mud in Australia and has been trialed on other tailings, including on coal tailings and fly ash in Australia, and oil sands tailings in Canada.

A scroller, as shown in Figure 19a, has a very low bearing pressure of 3 to 5 kPa and is used first. The principles of tailings or residue farming by scroller are as follows:

- The tailings or residue is poured to a thickness of 700 to 900 mm, up to three times the thickness to which surface desiccation would extend.
- The first pass of the scroller is applied after some initial drying and strengthening of the tailings or residue surface, to allow safe and efficient operation.
- If the bearing pressure from the scroller and/or the tailings or residue surface is too soft, bogging of the scroller can occur.
- A scroller will only achieve minimal consolidation or compaction of the tailings or residue since its bearing pressure is low.
- A scroller should essentially "float" over the tailings or residue surface, creating trenches down the tailings or residue beach to facilitate the drainage of surface water, maximizing the tailings or residue surface area that is exposed to evaporation and strengthening, and exposing undesiccated tailings or residue after further farming.
- A scroller should not over-shear the tailings or residue by excessive or repeated farming, with about four scroller passes being optimal.
- The last scroller pass may be perpendicular to the beach, to smooth the surface prior to subsequent dozing since there should no longer be any surface water to drain.

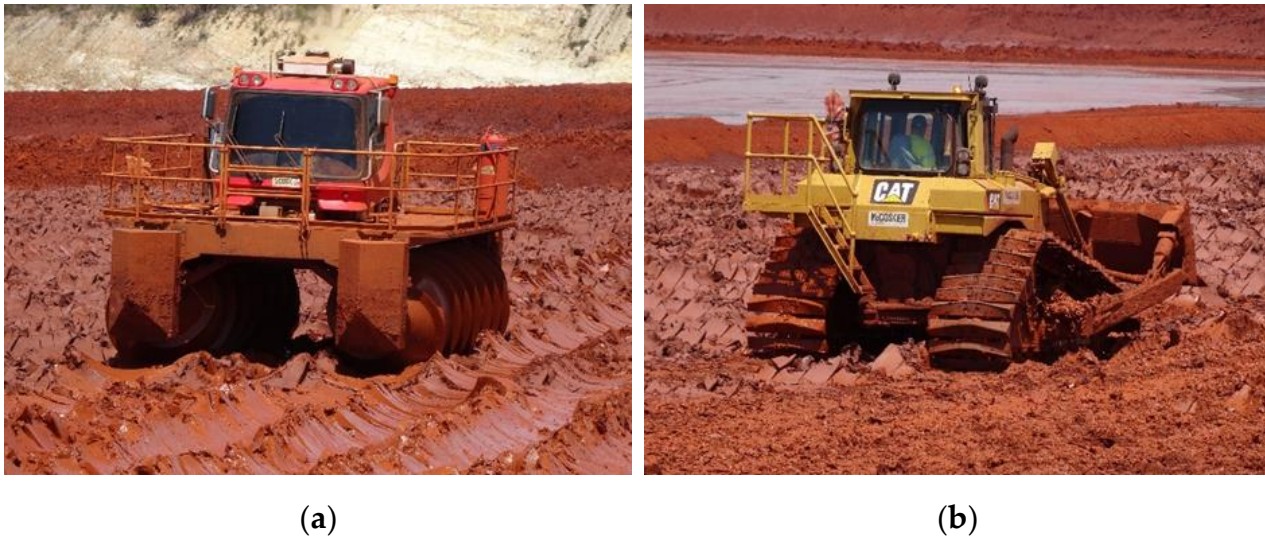

|  |  |
|:---:|:---:|
| (**a**) | (**b**) |

**Figure 19.** Farming red mud, by (**a**) a scroller, and (**b**) bulldozing with a D6 Swamp Dozer.

A D6 Swamp Dozer has a bearing pressure of about 35 kPa and can be applied once farming of the tailings or residue by scrolling has achieved sufficient shear strength and bearing capacity to safely support it, as shown in Figure 19b. The initial dozer pass, with the blade not in contact, serves to further remove any near-surface moisture. Dozing improves the already desiccated tailings or residue by compaction, leading to a further increase in dry density and shear strength. Typically, four to six dozer passes are applied.

Red mud typically has a specific gravity of about 3.0 and is difficult to densify, due to its forming, on slurry deposition, a loose "house of cards" structure of low permeability. Without farming, the dry density achieved is typically limited to about 0.7 t.m$^{-3}$, and desiccation is limited to a depth of about 300 mm. Scrolling can increase the dry density to about 0.9 t.m$^{-3}$, and dozing can increase it further to 1.3 to 1.4 t.m$^{-3}$. The cost of red mud farming is offset by the increased utilization of the available storage, by facilitating upstream raising (potentially using compacted red mud), and rehabilitation on closure.

*6.7. Paste Tailings*

Paste thickeners can raise the percentage of solids by mass to between 45% (for red mud) and 75% for metalliferous tailings. Paste tailings can readily be delivered under gravity as underground backfill (usually with cement added), or in-pit if the dewatering facility is located close to the discharge point, possibly on a mobile skid.

Underground tailings paste backfill will generally reach its intended destination under gravity, provided that the angle between the discharge and final points is steeper than 45°. Paste tailings disposal to a surface facility has had a relatively limited take-up due to the high cost of pumping and discharge management, with Bulyanhulu in Tanzania, Africa the most well-known application.

The overall tailings water recovery, as a percentage of the total water used in processing, increases to about 80% for tailings disposal as a high slump paste, and to 85 to 90% for a low slump paste, although the cost of paste production is high.

*6.8. Filtration and Dry Stacking*

The difference in consistency between wet and dry tailings filter cake is illustrated in Figure 20. The wet tailings filter cake is near-saturated and has the potential to flow on disposal, while dry tailings filter cake has a stress-induced "structure" and tends to remain intact on disposal.

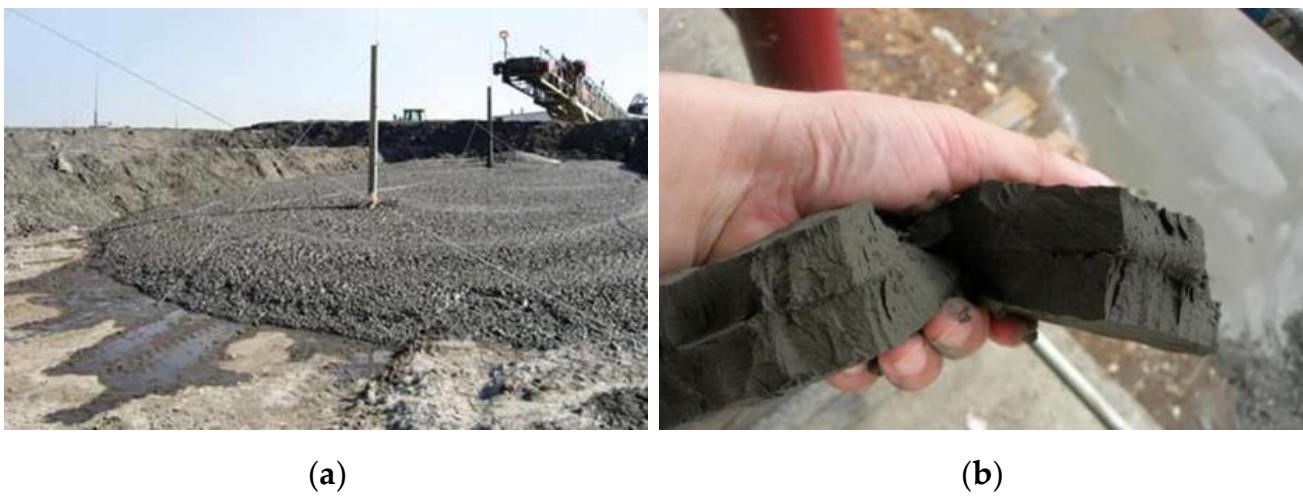

(**a**)                                       (**b**)

**Figure 20.** Consistency of (**a**) centrifuged wet tailings filter cake, and (**b**) filtered dry tailings filter cake.

Tailings filtration is best achieved under high pressure (1600 to 2100 kPa), with a cycle time of up to about 20 min. Although termed "dry", filtered tailings retain moisture and are more correctly described as "unsaturated". Dry tailings filter cake may be deposited in a stack, although compaction may be required for geotechnical stability, to prevent potential liquefaction, and to limit oxygen ingress and rainfall infiltration into potentially contaminating tailings, in order to minimize contaminated seepage.

Dry stacking has found most favor in dry climates such as in northern Chile, at La Coipa (Figure 21) [29] and other sites, and in southern Peru, to maximize water recovery in this region of limited freshwater supply. The cost of tailings filtration in the Atacama Desert region is comparable to the cost of seawater desalination. However, the dry stacking of filtered tailings has also been applied in wetter and colder climates, including at Greens Creek Metal Mine in Alaska [30] and at CSN's Casa de Pedra Iron Ore Mine at Congonhas, Minas Gerais, Brazil. The high cost of filtration has, to date, limited its application to tailings production rates of less than 20,000 dry tpd, although this limit is rising.

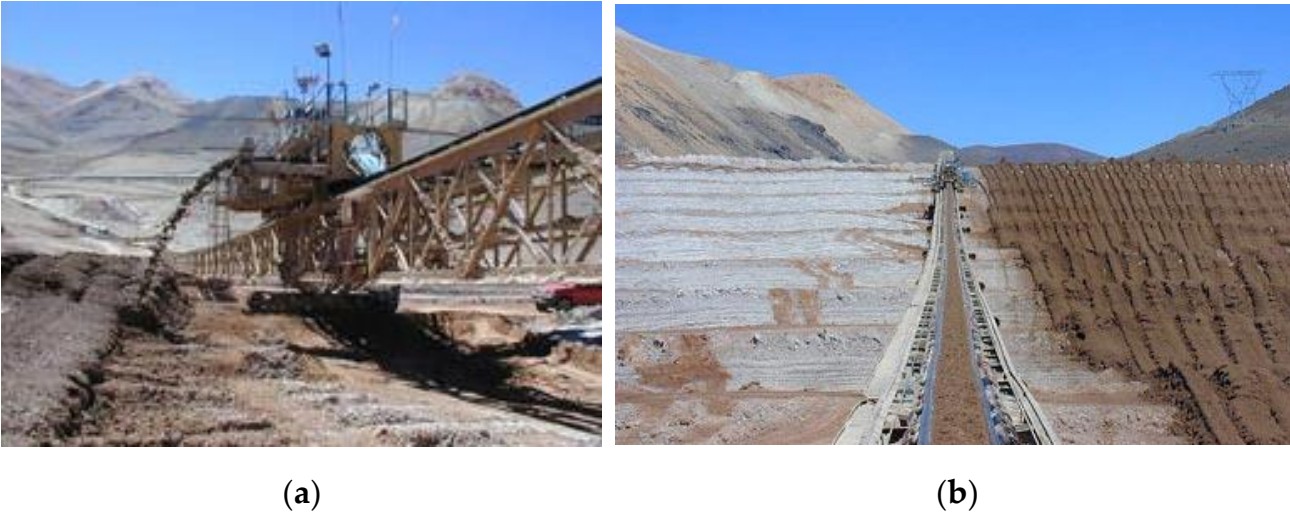

(**a**)                                       (**b**)

**Figure 21.** Dry stacking of filtered tailings at La Coipa, Atacama region, northern Chile, showing (**a**) stacker, and (**b**) overall stack [29].

With increasing the mechanical dewatering of tailings, costs increase dramatically. Thickening is about 1.5 times more costly than conventional slurry tailings, high-density thickening about 3 times more costly, paste about 8 times more costly, and filtration about

27 times more costly than slurry tailings [31]. However, this cost comparison ignores cost reductions due to increased process water recovery, the reduced storage volume required with increasing tailings dewatering, the consequently reduced containment required, and the reduced cost of rehabilitation and enhanced potential post-closure land use.

A more comprehensive cost comparison between slurry tailings deposition behind a dam, and tailings filtration and dry stacking with compaction, is provided in [32]. Tailings filtration and dry stacking with compaction indicated initial and average capital expenditure savings of 38% and 64%, respectively (almost 4 times the operating expenditure), with an average total cost saving of 33% over slurry tailings deposition behind a dam. The main cost of slurry tailings deposition behind a dam is the cost of dam raising, exacerbated by the low settled density and hence the high storage volume required for slurry tailings. In addition to the 80% saving in storage volume, the filtered tailings would also: recover about twice as much water by mass, plus metals and process chemicals; reduce seepage dramatically; be stable under seismic loading; be readily rehabilitated progressively and to a higher level of future land use. This cost comparison did not include closure costs.

*6.9. Co-Disposal of Tailings and Coarse-Grained Waste*

In British Columbia, Canada, coal tailings and coarse reject have been trucked and compacted in stable stacks that resemble a conventional dump rather than a tailings facility.

Since it was first trialed and introduced at Jeebropilly Coal Mine in the Ipswich Coalfields of South-East Queensland, Australia in 1990 [33], the combined coal tailings and coarse coal reject have been cost-effectively co-disposed by pumping at numerous coal mines in Australia and Indonesia. Pumped co-disposal in-pit at Jeebropilly Coal Mine is shown in Figure 22. In order to avoid pipeline blockages, the combined washery wastes are pumped at a low solids concentration of 25 to 30% by mass and at high velocity (up to 4 m.s$^{-1}$). While a steep upper coarse-grained beach (with about a 1-in-10 slope) is formed, the low solids concentration and high velocity result in the segregation of most of the fines and the generation of an undesirable flat (at about a 1-in-100 slope) fines beach (mostly tailings) beyond the upper beach. In addition, the inclusion of the coarse reject in a low-density medium results in high pump and pipeline wear.

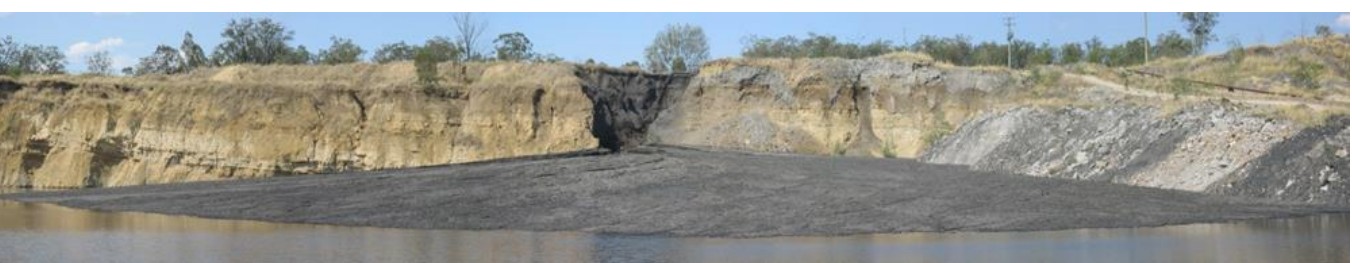

**Figure 22.** Pumped co-disposal in-pit at Jeebropilly Coal Mine.

The co-deposition in-pit of waste rock by end-dumping and thickening tailings by gravity has been practiced very cost-effectively at a number of sites, where completed pits have become available. At Kidston Gold Mines in north Queensland, Australia, the tailings and waste rock streams from the second pit were co-deposited entirely within the completed first pit [34], as shown in Figure 23. The pit was 240 m deep, and the waste rock was end-dumped from one side of the pit, while thickened tailings were deposited by gravity from a mobile thickener on a skid from the other side of the pit. The initial angle of repose of the waste rock slopes of 40° was flattened slightly to 38° as the waste rock settled, about 1.5% of its height or about 4 m, and extended at the "toe" by about 10 m.

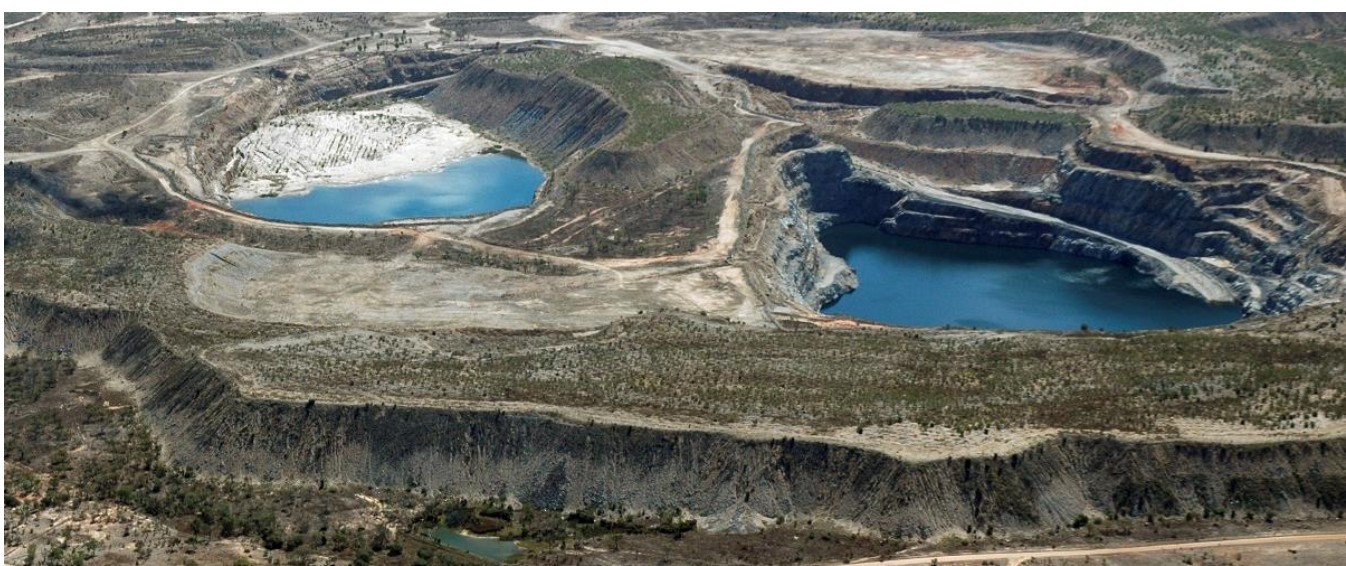

**Figure 23.** In-pit co-deposition of waste rock and thickened tailings (left-hand pit) at Kidston Gold Mines in north Queensland, Australia.

Dewatered tailings are also co-deposited in spoil dumps at coal mines in the Queensland Bowen Basin and New South Wales Hunter Valley Coalfields in Australia. The coal tailings are dewatered by belt-press filtration or centrifuging and are deposited from trucks in the coarse waste dumps, either in dedicated cells where the tailings remain flowable, or by end-dumping over a tip-head where the tailings are more soil-like. The main determinate as to whether the dewatered tailings are flowable is the clay mineral content and type of the coal seams being washed. A small percentage of smectite clay can render the tailings very difficult to dewater by any means.

*6.10. Integrated Waste Landforms*

Integrated waste landforms are increasingly being planned and employed in Australia, particularly at coal and iron ore projects, and elsewhere worldwide. This involves either the construction of a robust containment for thickened tailings using waste rock (including a low permeability core and/or drainage layers, as required) or the co-disposal of mixtures of filtered tailings and waste rock or coarse-grained processing wastes. The mixtures are delivered by combined pumping for coal washery wastes, or by haul truck or conveyor. This approach has also been employed in the wet tropics to encapsulate tailings and waste rock that are potentially acid-forming, placed behind a robust containment of more benign waste rock, created by paddock-dumping and compacted in layers.

*6.11. Tailings Reprocessing and Reuse and Reduced Tailings Production*

There is a long history of reprocessing gold tailings, sometimes more than once, such as in Johannesburg in South Africa and Kalgoorlie in Western Australia. What is relatively new is the water monitoring of base metal tailings for reprocessing, such as at Century Zinc Mine in North Queensland, Australia, as shown in Figure 24. Reprocessing tailings at Century has made it the world's tenth-largest zinc producer (formerly the third largest). The residual tailings are deposited into the completed pit, with the potential to completely remove the environmental liability of the surface tailings facility.

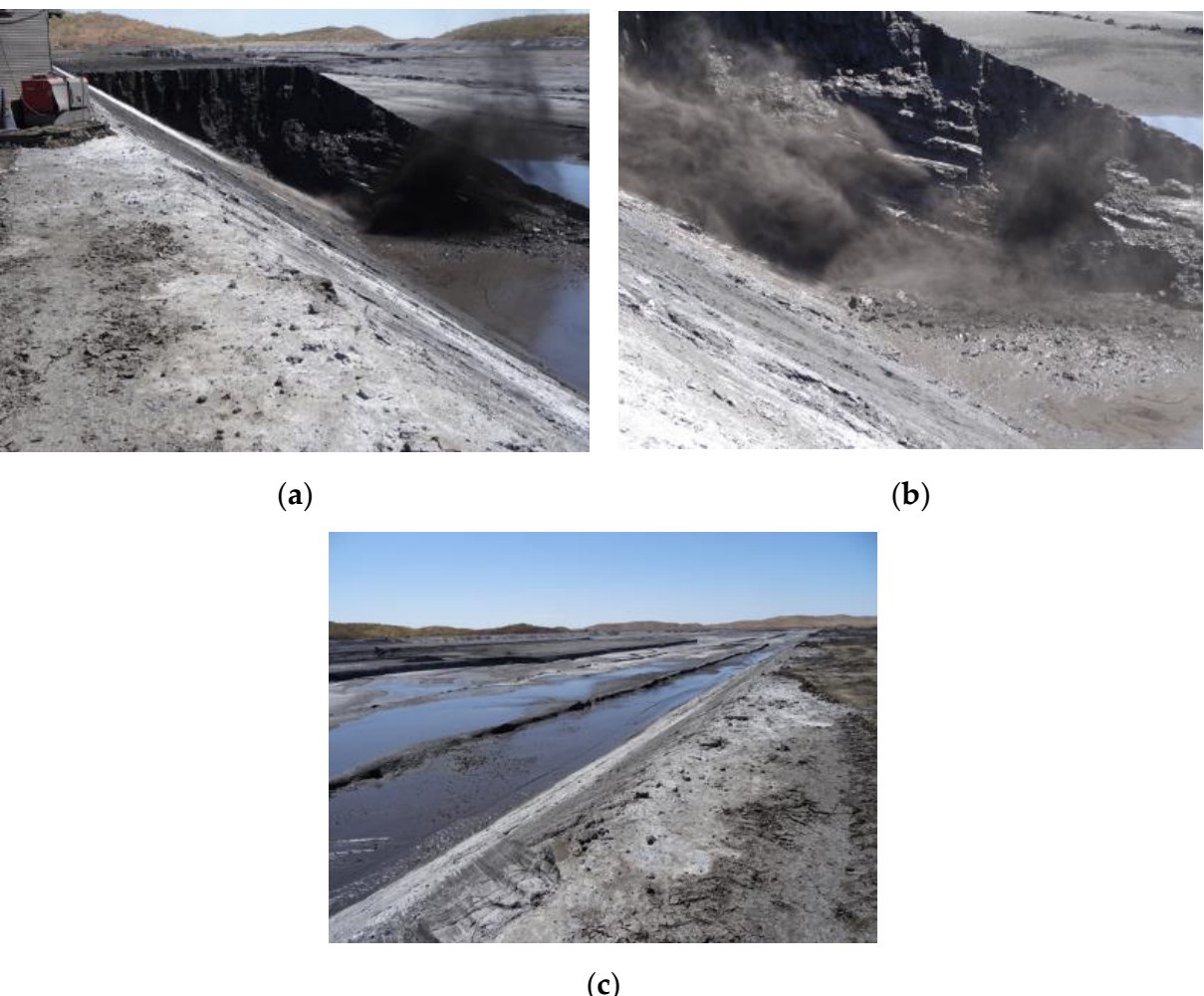

**Figure 24.** Water monitoring of tailings for reprocessing at Century Zinc Mine, showing: (**a**) water monitor, (**b**) re-slurried tailings, and (**c**) flowing slurry tailings toward a sump for recovery.

Examples of tailings reuse include for bricks and other building products, the use of pozzolanic power-station fly ash as a partial replacement for cement, and for the purposes of geopolymerization, as an environmentally friendly and low carbon replacement for natural sources [35,36].

In response to the ever-increasing production of tailings because of decreasing ore grades and increasing demand for minerals, attention is now being paid to finding ways of reducing tailings production. Another driver has been the rising cost of energy and other mining and processing costs. The primary focus has been on coarse particle and/or dry processing.

### 6.12. PasteRock^TM and GeoWaste^TM

The co-disposal of filtered tailings and waste rock can be achieved by mechanical mixing to form "PasteRock^TM", patented by Golder Associates, which has been trialed in Papua New Guinea, and in Canada for mine waste covers [37]. More recently, Goldcorp patented "GeoWaste^TM", which incorporates filtered tailings, combined with screened or crushed waste rock [38].

The practical and economic challenges that must be overcome to promote the combination of filtered tailings and waste rock include:

- Minimizing the extent to which the tailings must be dewatered, to save costs, while not compromising the stability of the combined tailings and waste rock.

- Minimizing the crushing or screening of the waste rock to allow mixing with the filtered tailings and transportation. The largest size of the waste rock for conveying is about 200 to 300 mm, while run-of-mine waste rock can be trucked.
- Achieving adequate mixing of the filtered tailings and waste rock. This is unlikely to occur on a conveyor since the two waste streams tend to remain separate. It is also unlikely to occur upon dumping from a haul truck since the coarse fraction tends to ravel further. However, mixing may be achieved by placing a number of drop points into hoppers along a conveyor line. It may also be achieved by placing the coarse-grained waste rock in the base of a truck, with the filtered tailings over the top to the designed ratio, to avoid the filtered tailings sticking to the truck tray and to force mixing on dumping.
- Compaction of the mixture may be required to produce a stable deposit, although compaction could be restricted to the perimeter of the emplacement.

The benefits of combining filtered tailings and waste rock can include an increased shear strength, reduced compressibility, and permeability that is lower than that of the waste rock alone but is higher than that of tailings alone.

*6.13. Briquetting of Tailings*

Briquetting involves forcing a slurry between two rollers under very high stress (of the order of 50 MPa), and has been shown to be very effective in dewatering ultra-fine black product coal [39]. In this study, the ultra-fine product coal that was initially at 40 to 45% total moisture content (mass of water/total mass, expressed as a percentage) was dewatered to briquettes of about 15% total moisture content (85% solids by mass). The very high stresses imposed over a very limited time duration resulted in further dewatering of the briquettes in a dry atmosphere, to about 2 to 5% total moisture content (95% solids). The air-dried briquettes can re-wet in a humid atmosphere, but only to about 15% total moisture content, and they retain their "briquette" structure. However, the very high initial capital and operating costs of briquetting, and scale-up issues, would discourage its application to tailings.

*6.14. Barriers to the Implementation of Innovative Tailings Management*

The conventional disposal of slurry tailings remains the preferred choice for many mine sites. Barriers to the implementation of the wide range of innovative tailings management options available include:

- The continued use of NPV accounting with a high discount factor (typically 6 to 10%, which is three to five times the consumer price index). This approach favors tailings management options that are less costly (particularly in capital expenditure terms) in the short term, delaying long-term expenditure and rehabilitation. Favoring low capital expenditure can come at the expense of increasing operating expenditure, and is likely to exacerbate undesirable outcomes and blow-out rehabilitation costs.
- The mechanical dewatering of tailings, and the co-disposal of tailings and waste rock, are seen as too costly in a narrow comparison with conventional tailings transport and disposal as a thickened slurry, a view that is reinforced by NPV accounting.
- There are perceived and real technical difficulties associated with the mechanical dewatering of tailings and the co-disposal of tailings and waste rock. Very fine-grained tailings and tailings with a high clay mineral content are difficult to thicken and filter, and it is difficult to mix tailings and coarse-grained waste.
- There is an underlying inherent resistance to change and doing something other than what has always been done, often disguised as unsubstantiated claims about perceived high costs, perceived technical obstacles, and perceived uncertainty.
- The uncertainty, and hence the perceived higher risk, attached to new approaches also serve to discourage innovation.

## 7. Closure and Rehabilitation of Tailings Facilities

The closure and rehabilitation of tailings facilities should be considered early on in their planning, and continuously during operation, since options decrease and costs increase during the mine's life, as highlighted schematically in Figure 25 (taken from the *Global Acid Rock Drainage* (GARD) *Guide* [40]).

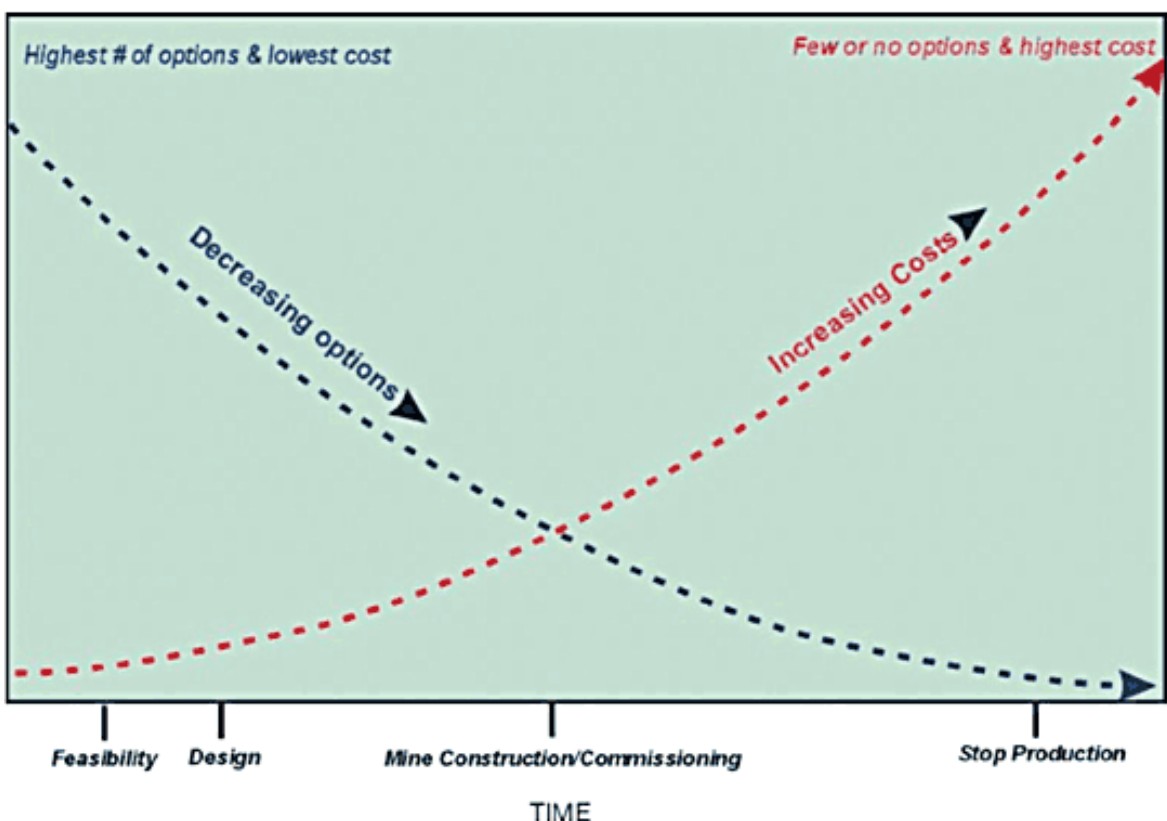

**Figure 25.** Choice of cover type, related to climate [40].

The aim of the closure of tailings facilities is to leave safe, stable and non-polluting structures in perpetuity, with stakeholder input to address the agreed post-mining land use and/or ecological function post-closure. Irrespective of the post-closure use(s) that are agreed upon, considerations for tailings facilities entering their closure phase should include:

- Geotechnical stability: the aim should be to convert tailings dams to be stable land-forms, removing credible failure modes. This may or may not require slope flattening or buttressing. With the cessation of tailings deposition, the daily water input to the tailings facility will generally be dramatically reduced, and the tailings would generally be expected to drain down. This would render the dam more geotechnically stable. However, the tailings may be recharged by high rainfall (in the absence of a spillway), and may not drain down in a wet climate, leading to no improvement in geotechnical stability.
- Erosional stability: erosion may be exacerbated by a lack of revegetation on the tailings cover and dam slope and/or a fine-grained soil surface texture.
- Differential tailings settlement: differential settlement will arise due to variable tailings depth and consistency, potentially affecting the slope profile and drainage.
- Geochemical stability: Poor water quality (such as saline, and/or acidic, or alkaline water) may develop, after a lag:
  - In ponded water on a surface of the tailings facility;
  - In seepage emerging at low points around the toe of a surface facility;

    ○  On seepage infiltrating to any groundwater resource beneath a tailings facility; or

    ○  In a pit backfilled with potentially contaminating tailings, particularly if it could become a source of contaminated seepage or runoff.

Wet and soft tailings resulting from conventional slurry tailings deposition are difficult and expensive to physically cover and rehabilitate, particularly at the end of the mine life, when the mine is no longer producing revenue and construction equipment is being demobilized. Furthermore, such tailings will limit the future land-use potential of the facility. On the other hand, their high degree of saturation will limit the oxidation of any sulfides present in the tailings, reducing the potential for contaminated seepage.

*7.1. Covering Tailings*

The rehabilitation of tailings can vary, including benign water covers in wet climates, the direct revegetation of benign tailings, and soil covers, particularly in dry climates. As shown in Figure 26, the GARD guide [40] recommends that the choice between cover types is based on climatic conditions, guided by:

- Water covers: appropriate in wet climates as effective oxygen barriers, provided that they are a nominal 2-m deep to cater for wave action and potential mobilization of tailings, are retained by stable dams, and have an adequate supply of rainfall-runoff to maintain them.
- Water-shedding soil covers: appropriate in moist climates to promote revegetation for erosion protection, and rainfall runoff to limit net percolation of rainfall.
- Store and release soil covers that store wet season rainfall, releasing it through evapotranspiration during the dry season; appropriate for dry or seasonally dry climates to sustain revegetation, limit net percolation of rainfall, and prevent erosion through preventing rainfall-runoff during heavy rainfall events.

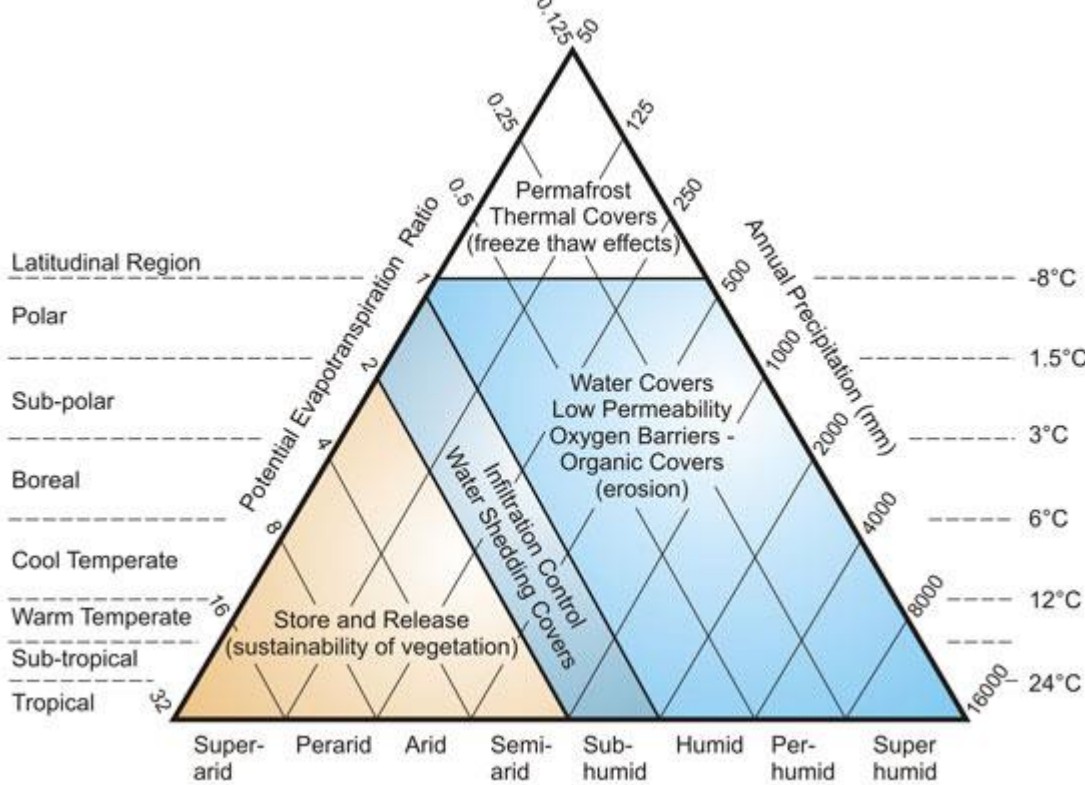

**Figure 26.** Choice of cover type related to climate [40].

In seasonally dry climates, store and release covers are more robust than rainfall-shedding covers since they better sustain revegetation, due to their greater rainfall infiltration and water storage capacity, and since they limit erosion by preventing rainfall runoff. Store and release covers require a base sealing layer (or a desiccated, hardpan, or compacted tailings surface, if suitable) to limit the breakthrough of rainfall infiltration into the underlying tailings, and may take advantage of the natural tailings beach slope to direct clean excess rainfall infiltration toward a collection point, limiting breakthrough into the underlying tailings.

### 7.2. Soil Cover Designs

The GARD guide [40] also provides schematics of soil cover designs, as shown in Figure 27, which increase in complexity, construction difficulties, potential performance and cost from left to right. The implication from Figure 27 is that the more complex and thick the cover, the more effective it will be. However, covers should be appropriate to the site conditions, in particular the climate, and more complex and thicker covers may not be the most effective choice for a given site, just more expensive. The schematics have generated much confusion, particularly in Australia. The Base Case is a single growth medium layer. Cover I, indicating a thicker single layer of growth medium than the Base Case, is inferred to be "better" than the base case. However, thicker is not necessarily better. A thick growth medium can lead to the infiltration of rainfall to a depth that makes it inaccessible to revegetation. This can also lead to increased net percolation into the underlying tailings, producing a result that may be worse than having no cover at all, since the cover would constitute a "sponge" that would increase rainfall infiltration, compared with a desiccated, hardpan, or compacted tailings surface.

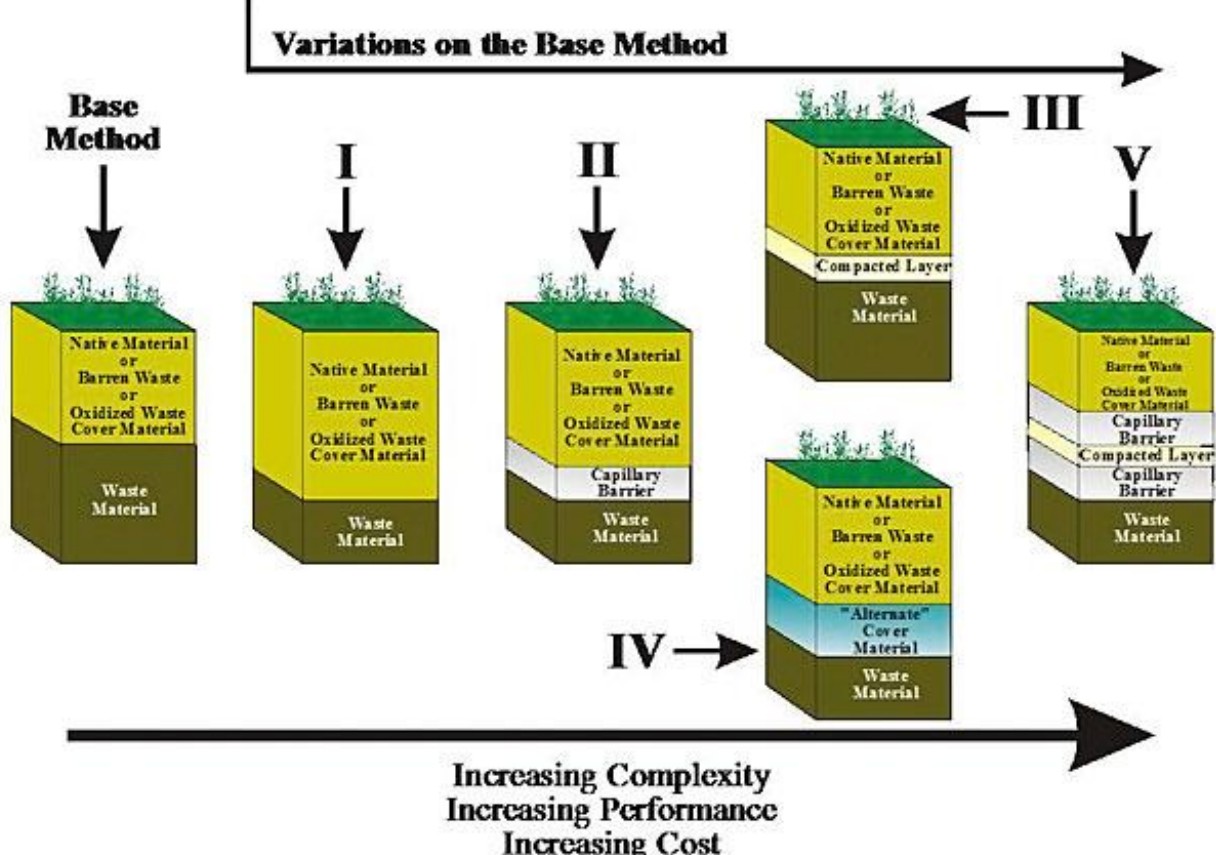

**Figure 27.** Various soil cover designs [40].

Cover II adds a capillary barrier (or break) beneath the growth medium, which may be desirable for limiting the uptake of any salts from the underlying tailings into the growth medium. A capillary barrier relies on a material that limits saturation and flow driven by capillary action and must be carefully selected and sized to ensure that it is effective and will remain so. One suitable capillary barrier material is clean gravel. Run-of-mine waste rock would likely not be suitable for use as a capillary barrier without the crushing of coarse-grained particles and screening to remove fines. A capillary barrier is a "drain", and so it should be overlain with a sealing layer to limit the net percolation of rainfall into the underlying tailings.

Clean gravel would require a thickness of greater than 200 mm to remain effective as a barrier. The capillary barrier thickness must allow for the possible infiltration of fines from the overlying growth medium, which would render it less effective over time. The particle size of the capillary barrier must be matched to that of the overlying growth medium, using filter criteria, to ensure that the infiltration of fines into the capillary barrier is limited by "arching" between particles. Finding suitable material for an effective capillary break, and ensuring its sustainability, are problematic.

Cover III adds a compacted (clayey) sealing layer beneath the growth medium, which is desirable, particularly for a store and release cover, to "hold up" rainfall infiltration within the overlying "rocky soil mulch" layer. Cover IV is a variation on Cover III, in which the compacted layer is replaced by an "alternative" sealing layer, such as a geomembrane, bituminous geomembrane, or geosynthetic clay liner (GCL). A sealing layer could also potentially be achieved by a desiccated, hardpan, or compacted tailings surface.

Cover V incorporates three layers separating the growth medium from the tailings, comprising a compacted fine-grained layer sandwiched between two capillary barriers. This cover was apparently based on the cover with capillary barrier effects (CCBE) applied at the Les Terrains Aurifères (LTA) mine site sulfide tailings impoundment, near Malartic, Abitibi, in Québec, Canada, in a net positive precipitation climate [41].

The CCBE, constructed on the LTA tailings in 1998, comprises 500 mm of sand (a capillary barrier) placed on the sulfidic tailings, overlain by 800 mm of fine-grained, non-acid-generating tailings (a moisture-retaining layer) that is in turn overlain by more than 300 mm of sand and gravel (protection and drainage layer). The design objective was to maintain a minimum degree of saturation of 85% in the moisture-retaining layer to effectively reduce the oxygen flux from the atmosphere to the underlying acid-generating tailings. Near-saturation of the moisture-retaining layer was to be maintained by a combination of enhanced rainfall infiltration and suction in the lower capillary barrier. The intention of the sand and gravel surface layer was to limit revegetation, so as not to reduce rainfall infiltration. The same cover was applied to the side slopes of the tailings impoundment (also comprising sulfidic tailings).

The cover initially functioned as intended on the top of the impoundment, which initially remained un-vegetated, but it was not so successful on the side slopes, due to gravity drainage. However, volunteer revegetation was established on the LTA CCBE, commencing the year after construction [42]. Eight functional groups of plants were identified, with herbaceous plants being the most abundant. Of the 11 tree species identified, the four most abundant were poplar, paper birch, black spruce and willow. Root excavation showed that tree roots penetrated the moisture-retaining layer, with an average root depth of 400 mm and a maximum root depth of 1.7 m.

After 10 years, although the LTA CCBE was effective in reducing the oxygen flux from the atmosphere to the acid-generating tailings, the quality of the seepage from the tailings impoundment still did not meet the Québec water quality standards, and dolomitic drains were constructed as a passive treatment [43].

The CCBE was designed for a specific purpose, in a net positive precipitation climate, and appears to be the only Cover Type V double capillary barrier cover applied in practice. Double capillary barrier (or break) covers have been promoted by some regulators in Australia, notably in Queensland. This is perplexing, since the majority of mine sites in

Queensland, and elsewhere in Australia, are in semi-arid to arid climates, for which the GARD guide recommends a store and release cover. Attention is best paid to the design, materials selection, and construction of such store and release covers.

### 7.3. Store and Release Cover Design

The key elements of a store and release cover [44,45], which was developed for seasonal, dry climates, are: (i) a thick, loose, rocky soil with a mulch growth medium layer that has an undulating surface to store the wet season rainfall without inducing runoff; (ii) an effective sealing layer at the base of the cover to hold-up rainfall infiltration; (iii) the appropriate choice of sustainable revegetation to release the stored rainfall during the wet season, through evapotranspiration.

The required thickness of rocky soil mulch growth medium will depend on the wet season rainfall pattern and the rooting depth of the vegetative cover applied and is typically up to 2-m thick. It is sized so as to accommodate within its available voids the majority of the wet season rainfall. The typical porosity of the loose, rocky soil mulch is about 0.25, providing up to 250 mm of storage volume per meter thickness of mulch.

Too thin a growth medium will not accommodate the wet season rainfall infiltration and will not support revegetation during the dry season. Too thick a growth medium could lead to rainfall infiltration beyond the reach of the revegetation. In a dry climate, store and release covers are more robust than rainfall-shedding covers, which have a more limited store and release capability and promote runoff and the potential for erosion.

The sealing layer should achieve a saturated hydraulic conductivity of less than $10^{-8}$ m/s (equivalent to a potential percolation rate of less than 300 mm/year, when water is available), so that in its usual unsaturated state in a dry climate its hydraulic conductivity will be less than perhaps $10^{-10}$ m/s (a potential percolation rate of less than 3 mm/year).

In the typically dry climate found over much of Australia, and in other dry regions, rainfall occurs on only perhaps 30 days/year, so that water may be available on top of the sealing layer for perhaps 10% of the time, reducing the potential percolation rate to less than 30 mm/year, or less than 5% of the typical average annual rainfall, similar to the typical natural percolation rate. High net percolation will be associated mainly with extreme rainfall events. A store and release cover should cycle annually between wet and dry states without a net wetting-up (which would lead to net percolation) or drying out (which would cause revegetation dieback and subsequent rainfall-induced erosion). An extreme rainfall event may wet up the cover, potentially leading to net percolation into the underlying tailings. In Australia's generally arid to semi-arid climate, a mixed eucalypt tree cover represents the only sustainable means of achieving the required evapotranspiration rates from a store and release cover that can handle extreme rainfall events and be sustainable in the long term. Australian eucalypts are quite adaptable to the climatic variability in the country, and their water uptake can vary by an order of magnitude, according to the availability of water.

There is ongoing concern and debate about the effect of tree roots on cover integrity on the tops of tailings facilities, from a number of standpoints. Tree roots can potentially penetrate through the cover thickness toward the underlying tailings. However, a desiccated, hardpan or compacted tailings surface is essentially a root barrier. This gives tree roots no encouragement to penetrate to depth in search of moisture or nutrients. Instead, the tree roots would grow laterally through the overlying cover. Since the rocky soil mulch is loose and granular, trees are unlikely to promote cracking and the development of preferred seepage pathways through the cover. The height of the trees will be limited by the climate, and by the thickness, water-holding and nutrient capacity of the rocky soil mulch, while their root patterns will not penetrate the sealing layer but will instead grow laterally. Sufficient rocky soil mulch thickness and limited tree height will lessen the possibility of wind blow-down and any possible threat to the integrity of the cover. Should wind blow-down of shrubs and trees occur, the limited rooting depth will limit the impact on the cover, and the coarse-grained rocky soil mulch will tend to self-heal.

Although store and release covers have gained popularity for mine wastes in dry climates, they have not always been well designed and constructed. Failings include the lack of suitable and sufficient cover materials and poor construction control. A further failing is that cover placement is necessarily delayed until the completion of filling of the tailings facility, by which time tailings that have the potential to generate poor-quality seepage are already doing so. Placing any cover on wet and soft conventional tailings is also a challenge. In response to the perceived poor performance of the store and release covers, a composite cover is seen by regulators to be "better", while operators see it as being more costly.

### 7.4. Treatment of the Side Slopes of a Surface Tailings Facility

The erosion of the side slopes of a surface tailings facility is a function of the surface texture and revegetation (Figure 28) [46]. A grass cover needs to extend over the majority of the surface of a slope to be effective in limiting erosion and, during the long dry season over much of Australia, and other dry regions, this cannot be relied upon. Further, extended droughts cause revegetation dieback. This has implications for the use of erodible, fine-grained topsoil on tailings facility side slopes. Erosion resistance under such climatic conditions requires a rocky surface texture on the slopes.

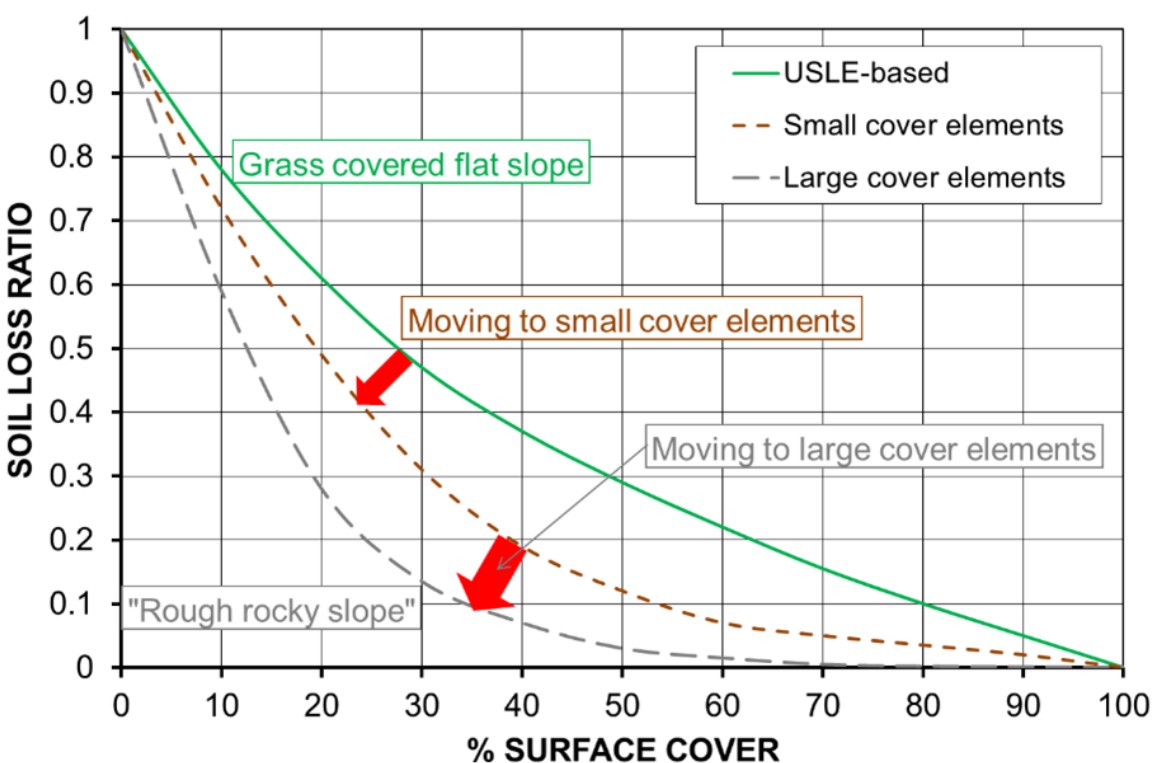

**Figure 28.** Soil loss as a function of the percentage of surface cover and surface texture [46].

### 7.5. Conventional Cost-Focused Rehabilitation versus Value-Added Rehabilitation

The conventional cost-focused approach by both operators and regulators to the rehabilitation of surface tailings storage is often at odds with the potential for value-added rehabilitation, as described in Table 1 [47]. A focus on the cost discourages and delays rehabilitation activities post-closure, which in turn is likely to lead to increased impacts over time, exacerbating the situation. In contrast, identifying and realizing potential opportunities for value-added rehabilitation and post-closure land use sets the rehabilitation budget and is a potential win for all stakeholders, including the operator, future land users and the government.

**Table 1.** Conventional cost-based rehabilitation versus value-added rehabilitation.

| **Conventional Cost-Based Rehabilitation** |
|---|
| Production rules |
| Rehabilitation is seen by the operator and regulator as a "cost" |
| The operator discounts cost over time, discouraging rehabilitation |
| Infrastructure such as power lines and buildings are stripped for little financial gain |
| Rehabilitation is limited to "smoothing" and "greening" (where sustainable) |
| Post-closure land use and function are limited |
| The operator is threatened with loss of financial and social licenses to operate |

| **Value-Added Rehabilitation** |
|---|
| Post-closure "value" is identified up-front |
| Examples of post-closure value include: |

- Reprocessing of tailings to extract metals of value, depositing the residual tailings in-pit and reducing the rehabilitation liability;
- Industrial land use;
- Renewable energy–solar, wind and pumped storage, delivered to the grid via mine transmission lines;
- Agriculture and/or fishery impoundment;
- Tourism and heritage (the older the better).

| |
|---|
| Value sets the rehabilitation budget |
| Potential wins for the operator, future land user and the government |

## 8. Discussion and Conclusions

The commonly held perception that transporting tailings as a slurry to a surface dam is the most economical solution has driven conventional slurry tailings disposal and needs to be challenged. The way in which tailings management is costed needs to move beyond relying largely on an NPV approach, with a high discount factor, to a more whole-of-life cost approach. Tailings management must also consider the nature of the tailings and how they may change through the life of the mine and, most importantly, the climatic, topographic and seismic settings of the mine. The site settings are not under the control of the operator, although they can and must be accommodated during operations. Post-closure, the site settings will dominate the long-term performance of a tailings facility.

The ongoing rate of tailings facility failures is unacceptable to both industry and society, and there is a need to restore lost confidence and trust in the industry's ability to safely manage tailings, the ability of their consultants and contractors to design and construct safe facilities, and the capability of regulators to oversee this.

Past tailings dam failures have highlighted that tailings dams that fail have marginal stability, and the technical causes of failures are reasonably well known. They include the inappropriate use in the past of upstream construction in highly seismic regions such as Chile, and in wet climates such as Brazil. The industry in Chile responded to this, following the fatal 1965 El Cobre tailings dam failure, and turned around tailings practice there to be among the safest worldwide. Since the 2019 Brumadinho tailings dam failure in Brazil's wet climate, the upstream construction of tailings dams is no longer allowed there, and a global tailings standard (GISTM) has been developed. In more suitable climatic, topographic and seismic settings, where required controls are in place, upstream construction continues to be successfully employed. Weak foundation layers have been identified as a potential risk to tailings dam stability, particularly since they are raised progressively.

Risk assessment and the ALARP approach are now routinely applied to tailings management, and past tailings dam failures have raised the ALARP bar to an extremely high level, comparable to that applied to nuclear power stations. This is well above that achieved by water dams and is much higher than that achieved by conventional slurry tailings facilities in the past.

The GISTM has the aspirational goal of zero harm to people and the environment from tailings facilities, and elevates accountability to the highest organizational levels,

with new requirements for independent oversight. It expects global transparency and disclosure to improve stakeholder understanding. It sets as the design basis an extreme consequence classification or the current classification, with an upgrade to "Extreme" maintained throughout the tailings facility's life-cycle, including at closure. Many active tailings facilities are not classified as having an "Extreme" consequence and may need to be re-classified and/or upgraded to meet "Extreme". A plethora of guides accompany and complement the GISTM. Compliance with the GISTM is required of ICMM members, and is being taken up across the industry. This is challenging in the face of a shortage of qualified and experienced tailings practitioners at all levels.

There is much scope for the further development and implementation of alternative tailings management technologies and innovations, which will change the overall mine plan, moving away from a siloed approach to mining, processing and waste management. The natural resistance to doing things differently from the way they are usually done, yet expecting a different outcome, needs to be challenged.

Tailings facilities can be built to a similar margin of safety to that of water dams, at a probability of failure of about $10^{-4}$. This would prevent many tailings facility failures, with the associated loss of life, damage to infrastructure, and environmental harm. It would also restore the industry's financial and social licenses to operate, earning and restoring greater control of their operations. The implementation of existing and new approaches to technologies in tailings management could help to eliminate the risks posed by some conventional tailings facilities, possibly removing the risks altogether. Such approaches and technologies include:

- Addressing both the geotechnical and geochemical stability of tailings, which can be in conflict;
- Optimizing in-plant dewatering of tailings, particularly by thickening or filtration, and also considering paste tailings, particularly for gravity disposal in underground mines or in completed pits;
- Optimizing in-facility dewatering of tailings, possibly including on-off cells, and farming of deposited tailings that settle and consolidate poorly;
- Optimizing coagulation, flocculation and, where necessary, secondary flocculation, particularly for very fine-grained and clay mineral-rich tailings;
- Dry stacking of filtered tailings;
- Co-disposal of tailings and coarse-grained wastes, including mixing filtered tailings and waste rock;
- In-pit tailings disposal, particularly if final pit lakes containing water of diminishing quality can be avoided by complete back-filling;
- Integrated waste landforms that recombine tailings and coarse-grained wastes;
- Tailings reprocessing and reduced tailings production through coarse or dry processing;
- Possible briquetting of tailings, although this is unlikely to be economic; and
- Value-added tailings rehabilitation post-closure.

As discussed in the paper, there are several barriers to the implementation of innovative tailings management where they are indicated by site-specific conditions, particularly where existing facilities are concerned. Change will be more readily achieved in new mining projects and, hence, change in tailings management for the minerals industry as a whole will necessarily be generational. Case studies of the successful implementation of alternative tailings management and closure approaches should be published to inform the industry, their consultants and contractors, regulators, and other stakeholders. By this means, confidence and trust in the industry's management and closure of tailings facilities can be restored.

**Funding:** This review received no external funding.

**Conflicts of Interest:** The author declares no conflict of interest.

### Nomenclature of Abbreviations and Terms

| | |
|---|---|
| ALARP | As low as reasonably practicable |
| ANCOLD | Australian National Committee on Large Dams |
| Board | Board of Directors |
| CCBE | Cover with capillary barrier effects |
| CDA | Canadian Dam Association |
| CEO | Chief Executive Officer |
| EOR | Engineer of Record |
| F | Frequency of occurrence or annual probability of up to N fatalities |
| GCL | Geosynthetic clay liner |
| GISTM | Global Industry Standard on Tailings Management |
| ICMM | International Council of Mining and Metals |
| ICOD | International Committee on Large Dams |
| ITRB | Independent Tailings Review Board |
| JCOLD | Japanese Commission on Large Dams |
| KPI | Key Performance Indicators |
| N | Magnitude of occurrence or number of fatalities |
| NPV | Net present value |
| Pre-consolidation pressure | Highest historical stress |
| PRI | Principles for Responsible Investment |
| RTFE | Responsible Tailings Facility Engineer |
| SANCOLD | South African National Committee on Large Dams |
| TPR | Third-Party Reviewer |
| UNEP | United Nations Environment Programme |

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
