# Peer review of "Lessons from Tailings Dam Failures—Where to Go from Here?"

_minerals, doi:10.3390/min11080853_

Round 1
Reviewer 1 Report
The manuscript provides an extensive analysis about different approaches to tailings management. Although the review article contains relatively few references, it provides a comprehensive source for anyone who is interested in this topic.
There are a number of minor issues that the authors can address to improve the review paper:
- The caption to Figure 1 is not clear.
- Suggest that the authors add a list of nomenclature to describe the abbreviations and terms used in the paper.
- The subsection "5.11. Tailings Reprocessing and Reuse and Reduced Tailings Production" is recommended to be expanded to include other tailings reuse approaches. For example, through geopolymerization (Krishna et al. Ceramics International (2021) Volume 47, Issue 13, 1 Pages 17826-17843 (DOI: https://doi.org/10.1016/j.ceramint.2021.03.180. Lazorenko et al. Process Safety and Environmental Protection (2021) Volume 147, Pages 559-577 (https://doi.org/10.1016/j.psep.2020.12.028).
- Manuscript has 46 references. Mostly these are various online sources. This is not necessarily a bad thing, but please consider carefully if whether it can be expanded by analyzing more scientific articles.
Author Response
The manuscript provides an extensive analysis about different approaches to tailings management. Although the review article contains relatively few references, it provides a comprehensive source for anyone who is interested in this topic.
There are a number of minor issues that the authors can address to improve the review paper:
- The caption to Figure 1 is not clear. – This has been corrected.
- Suggest that the authors add a list of nomenclature to describe the abbreviations and terms used in the paper. – Done.
- The subsection "6.11. Tailings Reprocessing and Reuse and Reduced Tailings Production" is recommended to be expanded to include other tailings reuse approaches. For example, through geopolymerization (Krishna et al. Ceramics International (2021) Volume 47, Issue 13, 1 Pages 17826-17843 (DOI: https://doi.org/10.1016/j.ceramint.2021.03.180. Lazorenko et al. Process Safety and Environmental Protection (2021) Volume 147, Pages 559-577 (https://doi.org/10.1016/j.psep.2020.12.028). – Added.
- Manuscript has 46 references. Mostly these are various online sources. This is not necessarily a bad thing, but please consider carefully if whether it can be expanded by analyzing more scientific articles. – This has been given careful consideration, and there are now 48 references.
Submission Date
13 July 2021
Date of this review
20 Jul 2021 11:38:14
Reviewer 2 Report
Your article discusses a very important issues. It is crucial to have operating dams for the mining operations as every failure will cause a big negative impact on public acceptance. And naturally it is needed to keep the environmental effects in minimum.
I suggest that you discuss the general tailings management and challenges and aims of it more in details in your introduction, which is at the moment short.
You should have same font size in Figures; especially Figure 10, it’s axis’ names. Also check the rest of figures; there seems to be little difference in them. Figure 16 also needs special attention on the format. In part of your figures there is lines around them and in some there is not; you should pick one method and use it in all figures.
You give a nice series of different alternative tailings management methods. Very nice indeed. Maybe you could discuss the pricing a bit; investments and operating costs, so that the reader could have an understanding what kinds of amounts of money is there in question, if using different methods.
Your article is very nicely written and it gives a good throughout understanding on the methods available for dam management.
Author Response
Your article discusses a very important issues. It is crucial to have operating dams for the mining operations as every failure will cause a big negative impact on public acceptance. And naturally it is needed to keep the environmental effects in minimum.
I suggest that you discuss the general tailings management and challenges and aims of it more in details in your introduction, which is at the moment short. – The introduction has been expanded.
You should have same font size in Figures; especially Figure 10, it’s axis’ names. Also check the rest of figures; there seems to be little difference in them. Figure 16 also needs special attention on the format. In part of your figures there is lines around them and in some there is not; you should pick one method and use it in all figures. – All figures have been checked and modified as required.
You give a nice series of different alternative tailings management methods. Very nice indeed. Maybe you could discuss the pricing a bit; investments and operating costs, so that the reader could have an understanding what kinds of amounts of money is there in question, if using different methods. – Published cost data are very limited. Some cost data are provided for filtration and dry stacking in Section 6.8. Costs are mentioned in Sections 6.12, 6.13 and 6.14.
A sentence related to cost has been added to Section 6.2. A brief paragraph on the cost of flocculants has been added to Section 6.4. A sentence on the cost of operating on-off tailings cells has been added to Section 6.5. A sentence on the cost of farming has been added to Section 6.6. A brief comment on the cost of paste production has been added to Section 6.7. A brief mention of the cost-effectiveness of pumped co-disposal and co-deposition has been added to Section 6.9.
Your article is very nicely written and it gives a good throughout understanding on the methods available for dam management.
Submission Date
13 July 2021
Date of this review
19 Jul 2021 13:15:32
Reviewer 3 Report
In this manuscript, authors have prepared a review on “Lessons from tailings dam failures – Where to from here?”. It looks like a summary report since it lacks in-depth analysis and discussion. Moreover, the reviewer has many questions on the manuscript.
Comments:
Authors should clearly mention how the current manuscript meets the overall scope of the Minerals. Also, they should highlight the novelty of this review paper.
There is no reference in the introduction, sections 2.1, 2.2 and sections 3.1 and 3.2. Are the texts presented in these sections are your own views or general statement? If not, give proper citation to support the statements.
Fig. 1: Provide the source of information. Check all figures carefully and give the appropriate reference if the figure is not created by authors.
This manuscript contains various acronyms. The author can prepare a table to list all the acronyms.
Fig. 16: Explain all the acronyms in the figure legend.
Authors have reported various tailings management options. Among them, which would be a sustainable option. They should prepare a table by comparing advantages and disadvantages of various tailings management options.
Most of the figures present in this manuscript were directly taken from internet. Are these copyrighted materials?
Write a section on Future perspectives: Authors should list out what are the key knowledge gaps which should be considered in future to advance knowledge on the topic and development of sustainable technologies and better management strategies to address tailings management.
What are the critical factors (e.g., environmental, hydrological, and others) which impact the tailings dams failures?
References:
Authors have taken information from mostly internet/personnel communication. Authors are encouraged to make effort to use peer-reviewed articles to a maximum extent.
Author Response
In this manuscript, authors have prepared a review on “Lessons from tailings dam failures – Where to from here?”. It looks like a summary report since it lacks in-depth analysis and discussion. Moreover, the reviewer has many questions on the manuscript. – It has been accepted that this is a Review Paper.
Comments:
Authors should clearly mention how the current manuscript meets the overall scope of the Minerals. – The author leaves this to the Journal and Reviewers to judge, with three of the four reviewers favourable. Also, they should highlight the novelty of this review paper. – This is a Review Paper and, as acknowledged by the reviewers, who also acknowledge that it makes a valuable contribution to the topic of tailings management.
There is no reference in the introduction, sections 2.1, 2.2 and sections 3.1 and 3.2. Are the texts presented in these sections are your own views or general statement? If not, give proper citation to support the statements. – References are not needed in the Introduction or Sections 2.2 and 3.2, which are based on the author’s experience. References to the selected tailings dam failures are also now included in Section 3.1 (as well as in Sections 3.3 to 3.7).
Fig. 1: Provide the source of information. Check all figures carefully and give the appropriate reference if the figure is not created by authors. – The caption for Figure 1 has been added, and the source of the information is given in the text (as above). All figures have been checked carefully and appropriately referenced.
This manuscript contains various acronyms. The author can prepare a table to list all the acronyms. – This has been added as per the suggestion of Reviewer 1.
Fig. 16: Explain all the acronyms in the figure legend. – These are given in the text above Figure 16 and in the added Nomenclature of Abbreviations and Terms.
Authors have reported various tailings management options. Among them, which would be a sustainable option. They should prepare a table by comparing advantages and disadvantages of various tailings management options. – The author believes that the sections on each method go as far as they can in describing the advantages and disadvantages of each method. There is no single sustainable option, and they could only be compared on a multi-criteria basis for specific site conditions.
Most of the figures present in this manuscript were directly taken from internet. Are these copyrighted materials? – They are not copyrighted (sources listed as “may be subject to copyright” have not been used), and there is many photographs taken by the author or from public sources.
Write a section on Future perspectives: Authors should list out what are the key knowledge gaps which should be considered in future to advance knowledge on the topic and development of sustainable technologies and better management strategies to address tailings management. – This is well covered in the Discussion and Conclusions.
What are the critical factors (e.g., environmental, hydrological, and others) which impact the tailings dams failures? – This is covered in Section 3.2 and in the following sections that detail the various main causes of tailings dam failures.
References:
Authors have taken information from mostly internet/personnel communication. Authors are encouraged to make effort to use peer-reviewed articles to a maximum extent. – At 48 in number, the reference list is pretty comprehensive, and includes available print (24 in number) and mainly more recent internet sources (20 in number).
Submission Date
13 July 2021
Date of this review
24 Jul 2021 13:30:57
Reviewer 4 Report
Figure 10, format fonts should be harmonized according to the rest of the figures.
The manuscript gives a very and broad view of the tailings dam failures. It also brings the overall picture of alternative tailings management options.
Author Response
Figure 10, format fonts should be harmonized according to the rest of the figures. – Done as per the suggestion of Reviewer 1.
The manuscript gives a very and broad view of the tailings dam failures. It also brings the overall picture of alternative tailings management options.
Round 2
Reviewer 3 Report
No further comments.